# Glycolysis-dependent sulfur metabolism orchestrates morphological plasticity and virulence in fungi

**Dhrumi Shah[1,2], Nikita Rewatkar[1], Adishree M[1], Siddhi Gupta[1,2], Sudharsan Mathivathanan[1], Sayantani Biswas[1], Sriram Varahan[1]\***

[1]CSIR-Centre for Cellular and Molecular Biology, Hyderabad, India; [2]Academy of Scientific and Innovative Research (AcSIR), Ghaziabad, India

## eLife Assessment

This work identifies a novel, conserved link between glycolysis and sulfur metabolism that governs fungal morphogenesis and virulence. The **compelling** evidence, integrating multiple approaches, provides an **important** conceptual advance. A future mechanistic dissection of how sulfur metabolites interface with known pathways is encouraged.

**\*For correspondence:**
sriram.ccmb@csir.res.in

**Competing interest:** The authors declare that no competing interests exist.

**Abstract** Fungi exhibit remarkable morphological plasticity, which allows them to undergo reversible transitions between distinct cellular states in response to changes in their environment. This phenomenon, termed fungal morphogenesis, is critical for fungi to survive and colonize diverse ecological niches and establish infections in a variety of hosts. Despite significant advancements in the field with respect to understanding the gene regulatory networks that control these transitions, the metabolic determinants of fungal morphogenesis remain poorly characterized. In this study, we uncover a previously uncharacterized, conserved dependency between central carbon metabolism and de novo biosynthesis of sulfur-containing amino acids that is critical for fungal morphogenesis in two key fungal species. Using a multidisciplinary approach, we demonstrate that glycolytic flux is crucial to drive fungal morphogenesis in a cAMP-independent manner and perturbation of this pathway leads to a significant downregulation in the expression of genes involved in de novo biosynthesis of sulfur-containing amino acids. Remarkably, exogenous supplementation of sulfur-containing amino acids robustly rescues the morphogenesis defect induced by the perturbation of glycolysis in both *Saccharomyces cerevisiae* and *Candida albicans*, underscoring the pivotal role of de novo biosynthesis of sulfur-containing amino acids as a downstream effector of morphogenesis. Furthermore, a *C. albicans* mutant lacking the glycolytic enzyme, phosphofructokinase-1 (Pfk1), exhibited significantly reduced survival within murine macrophages and attenuated virulence in a murine model of systemic candidiasis. Overall, our work elucidates a previously uncharacterized coupling between glycolysis and sulfur metabolism that is critical for driving fungal morphogenesis, contributing to our understanding of this conserved phenomenon.

## Introduction

In response to changes in their environment, fungi exhibit remarkable morphological plasticity, allowing them to reversibly transition between specialized cell states. This inherent ability to rapidly adapt to the changes in their environment is essential for their survival and allows them to colonize diverse ecological niches, including various host organisms (*Gow et al., 2002*; *Lin et al., 2015*; *Naranjo-Ortiz and Gabaldón, 2019*). This conserved phenomenon of fungal morphogenesis

**eLife digest** Growing extra arms or tentacles, changing skin colour or texture – what sounds like the premise of a horror story is actually a common occurrence in nature. Many microscopic fungi can modify their shape to infect hosts and evade their immune defences. These organisms do not simply grow – they can alter their physical form depending on their environment. One moment, they are harmless, yeast-like cells, quietly living in the human gut; the next, they transform into long, thread-like filaments called hyphae that penetrate tissues and cause infections in animals and crops. Scientists call this extraordinary shape-shifting ability 'fungal morphogenesis'.

While scientists have identified the genes involved in this process, the role of cellular metabolism has been less well established. Previous research suggests that the presence of glucose is important for fungal morphogenesis. But how this sugar influences this shape-shifting behaviour was not completely understood. Understanding what truly controls this process could uncover new ways to prevent harmful fungi from causing diseases in plants and animals.

To determine whether glucose merely supports fungal morphogenesis or truly drives this change, Shah et al. used baker's yeast and a pathogenic yeast that causes thrush in humans. Both yeast species were grown under nitrogen-limited conditions, a known factor to trigger fungal morphogenesis. When glycolysis - the process that breaks down sugar molecules into energy - was blocked, neither species could shape-shift.

More detailed experiments showed that this was due to the need for glycolysis to produce sulfur-containing building blocks, such as amino acids, that support this transition. When these sulfur-containing building blocks were supplied from the outside, the fungi regained their ability to change shape – even under conditions compromising their ability to perform glycolysis.

This effect was observed in both yeast species, highlighting that this process must be conserved in potentially many yeast species. Blocking glycolysis disarms the pathogen, compromising its morphogenesis and rendering it vulnerable to host defence. This metabolic interference paves the way for a new class of precision antimicrobials that are both more potent in combating infections and less toxic to the infected host. Moreover, fungal morphogenesis has been implicated in causing resistance to existing antifungals. Therefore, targeting glycolysis will support the development of effective therapeutic strategies to increase the efficacy of existing antifungals.

encompasses a plethora of morphological forms, including pseudohyphae, hyphae, appressoria, and fruiting bodies to name a few (*Lin et al., 2015*; *Riquelme et al., 2018*). Importantly, these morphological transitions in various pathogenic fungal species significantly influence their interaction with the respective host, impacting their overall ability to cause persistent infections (*Min et al., 2020*; *Wang and Lin, 2012*). Fungal morphogenesis is conserved across a plethora of fungal species and has profound implications with respect to fungal ecology, anti-fungal drug resistance, and pathogenicity across a spectrum of hosts (*Christensen, 1989*; *Puumala et al., 2024*; *Sexton and Howlett, 2006*). Consequently, it has been a central focus of intense scientific investigation over the last several years (*Lin et al., 2015*; *Riquelme et al., 2018*). During dimorphic transitions in fungi, changes in cell shape is accompanied by changes in cell polarity, cell size, and cell-cell adhesion, and these changes are regulated by two major signaling pathways: the cAMP-PKA pathway and MAP kinase (MAPK) pathways. Both pathways are crucial for morphogenesis in fungal model systems, including *Saccharomyces cerevisiae* and *Candida albicans*. In *S. cerevisiae*, the cAMP-PKA pathway is a critical regulatory cascade that controls cellular differentiation, such as pseudohyphal growth, in response to glucose availability. The glucose signal is transduced through two parallel mechanisms: the G-protein-coupled receptor (GPCR) Gpr1-Gpa2 system and the intracellular Ras system. The activation of Ras is triggered by high glycolytic flux, which is directly controlled by the Hxt (Hexose Transporter) family dictating the precise rate of glucose import into the cell. Both the Gpr1-Gpa2 and Ras pathways activate the adenylate cyclase (Cyr1), leading to elevated intracellular cAMP levels. This cAMP binds to the regulatory subunit Bcy1, releasing and activating the catalytic Tpk subunits of Protein Kinase A (PKA). This PKA activation, driven primarily by Tpk2, promotes the shift to a filamentous morphology by inducing the expression of necessary genes required for pseudohyphal differentiation (*Colombo et al., 1998*; *Lorenz et al., 2000*; *Xue et al., 1998*). Similarly, this pathway drives the essential yeast

to hyphal transition in *C. albicans*, often via the regulator Efg1 (*Maidan et al., 2005*). Concurrently, the MAPK (Mitogen-Activated Protein Kinase) pathway operates in parallel to the cAMP-PKA cascade and coordinates fungal morphogenesis in response to extracellular stress and nutrient signals. In *S. cerevisiae*, the Kss1 cascade is the key effector that drives pseudohyphal growth, predominantly activating under nitrogen-limiting conditions (*Lorenz and Heitman, 1998*). In the opportunistic fungal pathogen *C. albicans*, the homologous Cek1 pathway is indispensable for coordinating the complex hyphal development. Cek1 acts as a multifaceted sensor, by processing external morphogenetic cues and often engaging directly with the cell wall integrity system to ensure structural stability during the rapid yeast to hyphal transition. This pathway is a critical determinant of virulence, as its activation promotes the expression of adherence factors and ensures the formation of biofilms, enabling the fungus to colonize host tissues and resist immune response (*Csank et al., 1998*). The cAMP-PKA and MAPK pathways function in a parallel manner, engaging in significant crosstalk to integrate diverse environmental signals and coordinate a precise morphological response (*Biswas et al., 2007*; *Kumar, 2021*). While significant progress has been made in elucidating the intricate gene regulatory networks that govern these morphological transitions (e.g. Genetic networks that orchestrate yeast to pseudohyphal differentiation in *S. cerevisiae* and yeast to hyphal differentiation in *C. albicans* have been well characterized [*Nantel et al., 2002*; *Ryan et al., 2012*]), a critical knowledge gap persists regarding the underlying metabolic determinants of fungal morphogenesis. Despite the recognized influence of nutrient availability on fungal morphogenesis (*Broach, 2012*; *Fleck et al., 2011*; *Kumar, 2021*), our understanding of the specific metabolic networks and their regulatory influence on morphogenetic switching in fungi remains incompletely characterized.

In this study, we employed *S. cerevisiae* and the human fungal pathogen, *C. albicans*, to dissect the metabolic underpinnings of fungal morphogenesis under nitrogen-limiting conditions, a known trigger for morphogenetic switching in these fungi (*Csank et al., 1998*; *Gimeno et al., 1992*). Our findings reveal a critical role for central carbon metabolism, particularly glycolysis, in facilitating the yeast to pseudohyphal transition in *S. cerevisiae* in a cAMP-PKA-independent manner and the yeast to hyphal transition in *C. albicans*, both of which have been shown to be crucial for their nutrient foraging ability in diverse niches and the ability of *C. albicans* to successfully infect a host (*Palková and Váchová, 2006*; *Thompson et al., 2011*). Comparative transcriptomic analysis identified a conserved metabolic network crucial for these morphogenetic switching events under nitrogen-limiting conditions. Importantly, we observed that perturbations in glycolytic flux using pharmacological and genetic means negatively impact the expression of genes involved in sulfur metabolism. Furthermore, our functional studies demonstrated that supplementation with sulfur-containing amino acids (cysteine or methionine) effectively rescued the morphogenetic defects resulting from glycolytic impairment, establishing a novel metabolic axis linking glycolysis and sulfur metabolism in the regulation of fungal morphogenesis. Finally, we investigated the implications of this glycolysis-dependent sulfur metabolism on the virulence of *C. albicans*. Our data demonstrate that a *C. albicans* mutant lacking genes that encode for a crucial glycolytic enzyme, phosphofructokinase-1 (Pfk1), exhibits reduced survival within macrophages and severely attenuated virulence in an in vivo murine model of systemic candidiasis, which is rescued in response to sulfur supplementation to the host.

Collectively, our research, for the first time, identifies a previously uncharacterized metabolic axis linking glycolysis and sulfur metabolism under conditions that induce fungal morphogenesis and underscores the importance of this metabolic network in regulating fungal morphogenesis across two key fungal species and the ability of the human fungal pathogen, *C. albicans*, to establish infection in a host. These findings address a fundamental knowledge gap in the field and contribute significantly to our understanding of a broadly conserved phenomenon that exists across a plethora of fungal species.

## Results

### Glycolysis is critical for pseudohyphal differentiation in a cAMP-PKA-independent manner in *S. cerevisiae*

Nitrogen limitation is essential for yeast to pseudohyphal transition in *S. cerevisiae* (*Gimeno et al., 1992*; *Pan et al., 2000*). However, the influence of carbon sources on this phenomenon is not completely understood. It has been observed that fermentable carbon sources (sucrose, maltose, glucose, etc.)

are important for inducing pseudohyphal differentiation under nitrogen-limiting conditions (*Van de Velde and Thevelein, 2008*). Conversely, when these were replaced with non-fermentable carbon sources, there was a significant reduction in pseudohyphal differentiation (*Strudwick et al., 2010*). This indicates that the presence of fermentable carbon sources is critical for pseudohyphal differentiation in *S. cerevisiae*. Crabtree-positive organisms like *S. cerevisiae* metabolize glucose primarily via glycolysis to meet their growth and energy demands (*Barford and Hall, 1979*; *De Deken, 1966*). In order to determine whether the ability of *S. cerevisiae* to metabolize glucose via glycolysis is critical for pseudohyphal differentiation under nitrogen-limiting conditions, we spotted *S. cerevisiae* wild-type Σ1278b on nitrogen-limiting media containing 2% (w/v) glucose (SLAD) with and without sub-inhibitory concentrations of 2-Deoxy-D-Glucose (2DG) and sodium citrate (NaCi), which are well-characterized glycolysis inhibitors (*Cramer and Woodward, 1952*; *Evans and Ratledge, 1985*; *Yoshino and Murakami, 1982*) and monitored pseudohyphal differentiation (*Figure 1A*). Our data demonstrate that sub-inhibitory concentrations of 2DG and NaCi significantly attenuated the ability of *S. cerevisiae* to undergo pseudohyphal differentiation (*Figure 1B*). We then isolated cells from these colonies and quantified the percentage of pseudohyphal cells (cells with a length/width ratio of 2 or more[*Schröder et al., 2000*]) in these colonies. Addition of 2DG or NaCi resulted in a significant reduction in pseudohyphal cells (*Figure 1C*) without compromising the overall growth of *S. cerevisiae* wild-type Σ1278b (*Figure 1G*). Another diploid *S. cerevisiae* strain that is known to undergo pseudo-hyphal differentiation under nitrogen-limiting conditions is the prototrophic CEN.PK strain (*Laxman and Tu, 2011*). In order to determine if our observations in Σ1278b can be recapitulated in CEN.PK, *S. cerevisiae* wild-type CEN.PK was spotted on SLAD with and without sub-inhibitory concentrations of 2DG or NaCi. Our data clearly demonstrate that 2DG and NaCi strongly inhibit pseudohyphal differentiation (*Figure 1B and C*) without compromising the overall growth of *S. cerevisiae* wild-type CEN.PK as well (*Figure 1G*).

Given that 2DG and NaCi exhibited a strong inhibitory effect on pseudohyphal differentiation under nitrogen-limiting conditions in a strain-independent manner, we wanted to corroborate these observations by using pertinent genetic knockout strains that are compromised in performing glycolysis in the presence of glucose (*Heinisch, 1986*; *Smith et al., 2004*). In order to do this, we generated knockout strains which lack the genes that encode for enzymes that are involved in the glycolysis pathway, including *pfk1* (phosphofructokinase-1) and *adh1* (alcohol dehydrogenase-1) (*Bennetzen and Hall, 1982*; *Foy and Bhattacharjee, 1978*; *Figure 1D*). Heinisch observed that a null mutant of *pfk1*, a key glycolytic enzyme, is viable and can grow on glucose as a sole carbon source under conditions where the cell performs aerobic glycolysis. However, this same mutant exhibits a significant growth defect when cultured anaerobically (*Heinisch, 1986*; *Lobo and Maitra, 1983*). Similarly, Dickinson et al. showed that a null mutant of *adh1* is viable. This is likely due to the presence of multiple alcohol dehydrogenase isozymes present in *S. cerevisiae* that likely compensate for the absence of *adh1* (*Dickinson et al., 2003*). The deletions of *pfk1* and *adh1* genes were confirmed through whole genome sequencing (*Figure 1—figure supplement 1A and B*). Our result demonstrates that the deletion of *pfk1* and *adh1* in the Σ1278b background (*ΔΔpfk1* refers to the *pfk1* knockout strain and *ΔΔadh1* refers to the *adh1* knockout strain) resulted in a significant reduction in pseudohyphal differentiation, compared to the corresponding wild-type strain, under nitrogen-limiting conditions (*Figure 1E*). We then isolated cells from these colonies and quantified the percentage of pseudohyphal cells in these colonies. Deletion of *pfk1* and *adh1* resulted in a significant reduction in pseudohyphal cells (*Figure 1F*), corroborating our data obtained using the glycolytic inhibitors, 2DG and NaCi. Similarly, deletion of *pfk1* and *adh1* in the CEN.PK background resulted in a significant reduction in pseudohyphal differentiation (*Figure 1E and F*). In order to rule out whether the reduction in pseudohyphal differentiation in these mutants is due to a general growth defect, we performed growth curve analysis wherein overnight grown cultures of the deletion strains (*ΔΔpfk1* and *ΔΔadh1)* along with wild-type strains (Σ1278b or CEN.PK) were diluted to $OD_{600}$=0.01 in fresh SLAD and allowed to grow at 30 °C for 24 hr. $OD_{600}$ was recorded at 3 hr intervals (*Figure 1H*). Our data suggest that the reduction in pseudohyphal differentiation is not due to general growth defect, as the mutants were able to grow similar to the wild-type strains in SLAD (*Figure 1H*). Taken together, our results clearly demonstrate that the ability of *S. cerevisiae* to efficiently metabolize glucose via glycolysis is critical for pseudohyphal differentiation under nitrogen-limiting conditions.

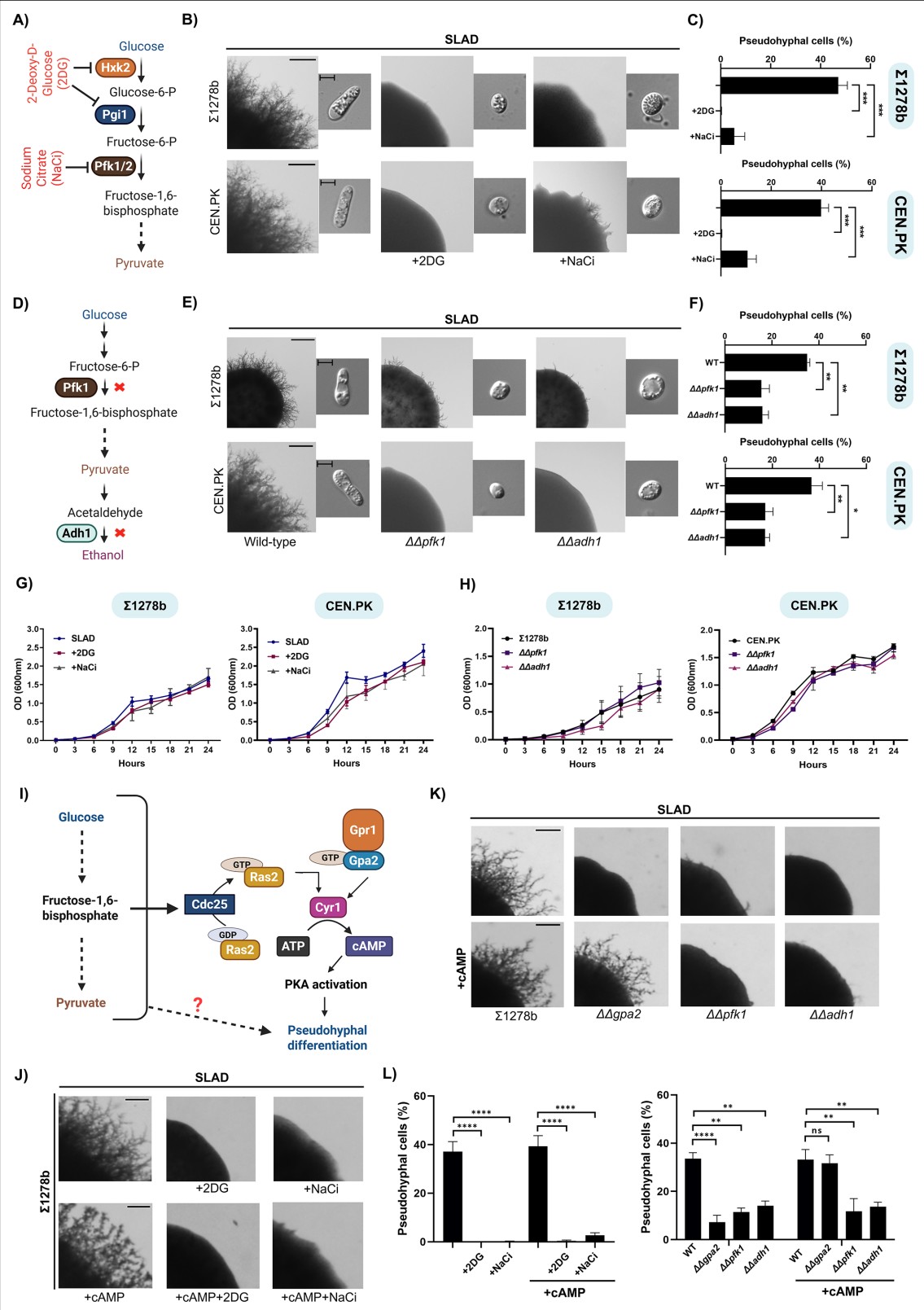

**Figure 1.** Glycolysis is critical for pseudohyphal differentiation in a cAMP-PKA-independent manner in *S. cerevisiae*. (**A**) Schematic overview of glycolysis and its inhibition by glycolysis inhibitors (2-Deoxy-D-Glucose (2DG) and sodium citrate (NaCi)). (**B**) Wild-type Σ1278b or CEN.PK were spotted on nitrogen-limiting medium containing 2% (w/v) glucose (SLAD) with and without sub-inhibitory concentrations of glycolysis inhibitors (2DG or NaCi), incubated for 10 days at 30 °C. Scale bar represents 1 mm for whole colony images and 5 μm for single cell images. (**C**) Cells from colonies were isolated

*Figure 1 continued on next page*

*Figure 1 continued*

and the length/width ratio of individual cells was measured using ImageJ and the percentage of pseudohyphal cells from the total population was determined. More than 500 cells were counted for each condition. Statistical analysis was done using one-way ANOVA test, ***($p<0.001$). Error bars represent SEM. (**D**) Schematic overview of glycolysis showing targeted gene deletions. (**E**) Pertinent knockout strains which lack genes encoding for enzymes involved in glycolysis, such as *pfk1* and *adh1* were spotted on SLAD along with wild-type Σ1278b or CEN.PK. Scale bar represents 1 mm for whole colony images and 5 μm for single cell images. (**F**) Cells from colonies were isolated and the length/width ratio of individual cells was measured using ImageJ and the percentage of pseudohyphal cells from the total population was determined. More than 500 cells were counted for each strain. Statistical analysis was done using one-way ANOVA test, **($p<0.01$) and *($p<0.05$). Error bars represent SEM. (**G**) Growth curve was performed to monitor overall growth of wild-type strains on SLAD, SLAD+2DG and SLAD+NaCi. Overnight grown culture of wild-type strains (Σ1278b or CEN.PK) were diluted to $OD_{600}=0.01$ in fresh SLAD medium with and without 2DG (0.05% (w/v)) or NaCi (0.5% (w/v)) and allowed to grow at 30 °C for 24 hr. $OD_{600}$ was recorded at 3 hr intervals. (**H**) Growth curve was performed to monitor overall growth of wild-type and knockout strains on SLAD. Overnight grown culture of deletion strains (ΔΔ*pfk1* or ΔΔ*adh1*) along with wild-type strains (Σ1278b or CEN.PK) were diluted to $OD_{600}=0.01$ in fresh SLAD and allowed to grow at 30 °C for 24 hr. $OD_{600}$ was recorded at 3 hr intervals. (**I**) Schematic overview of the role of glucose and glycolytic intermediates in the activation of the cAMP-PKA pathway during pseudohyphal differentiation. (**J**) Wild-type Σ1278b was spotted on SLAD containing sub-inhibitory concentration of 2DG or NaCi with and without cAMP (1 mM), incubated for 10 days at 30 °C. Scale bar represents 1 mm for whole colony images. (**K**) Pertinent knockout strains which lack genes that encode for enzymes involved in glycolysis, such as *pfk1* and *adh1* were spotted on SLAD with and without cAMP (1 mM), incubated for 10 days at 30 °C. ΔΔ*gpa2* strain was used as a control. Scale bar represents 1 mm for whole colony images. (**L**) Cells from colonies were isolated and the length/width ratio of individual cells was measured using ImageJ and the percentage of pseudohyphal cells from the total population was determined. More than 500 cells were counted for each condition. Statistical analysis was done using one-way ANOVA test, ****($p<0.0001$), **($p<0.01$) and ns (non-significant). Error bars represent SEM. This figure was created using Biorender.com.

The online version of this article includes the following figure supplement(s) for figure 1:

**Figure supplement 1.** Confirmation of phosphofructokinase-1 (*pfk1*) and alcohol dehydrogenase-1 (*adh1*) deletion in *S. cerevisiae*.

---

It is well-established that in *S. cerevisiae*, glucose acts as a key signalling molecule and is sensed by the G-protein-coupled receptor, Gpr1, that activates the cAMP-PKA pathway, which in turn is crucial for pseudohyphal differentiation under nitrogen-limiting conditions (*Lorenz and Heitman, 1997*; *Lorenz et al., 2000*). Additionally, a key glycolytic intermediate, fructose-1,6-bisphosphate (FBP), is also known to activate the cAMP-PKA pathway via activation of Ras proteins, further linking glucose metabolism to this crucial signalling cascade (*Peeters et al., 2017*; *Figure 1I*). To investigate whether the pseudohyphal differentiation defects that we observed due to glycolysis perturbation were solely due to the inability of glucose to activate the cAMP-PKA pathway, we performed cAMP add-back assays wherein *S. cerevisiae* wild-type Σ1278b was spotted on SLAD containing sub-inhibitory concentrations of 2DG or NaCi in the presence and absence of 1 mM cAMP (*Lorenz and Heitman, 1997*). Interestingly, the exogenous addition of cAMP failed to rescue pseudohyphal differentiation defect caused by the perturbation of glycolysis through 2DG or NaCi under nitrogen-limiting conditions (*Figure 1J and L*). We next asked if the exogenous addition of cAMP rescues the pseudohyphal differentiation defect exhibited by glycolysis-defective strains, ΔΔ*pfk1* and ΔΔ*adh1*. In order to do this, we performed similar cAMP add-back assays wherein *S. cerevisiae* wild-type Σ1278b along with ΔΔ*pfk1* and ΔΔ*adh1* strains were spotted on SLAD in the presence and absence of 1 mM cAMP. The ΔΔ*gpa2* strain was used as a control because it is well-established that the deletion of *gpa2* (G protein that activates cAMP production in response to glucose *Rolland et al., 2000*) impairs pseudohyphal differentiation and this defect can be fully rescued by the exogenous addition of cAMP under nitrogen-limiting conditions (*Harashima and Heitman, 2002*; *Lorenz and Heitman, 1997*). Corroborating our previous data, the exogenous addition of cAMP failed to rescue pseudohyphal differentiation defect caused by the perturbation of glycolysis via the deletion of *pfk1* and *adh1* but fully rescued the pseudohyphal differentiation defect exhibited by the ΔΔ*gpa2* strain (*Figure 1K and L*). Taken together, our results clearly demonstrate that the ability of *S. cerevisiae* to efficiently metabolize glucose via glycolysis is critical for pseudohyphal differentiation in a cAMP-PKA-independent manner under nitrogen-limiting conditions. This implies that glycolysis may be parallelly regulating other cellular processes essential for this morphological transition under nitrogen-limiting conditions.

## Comparative transcriptomics identifies glycolysis-dependent regulation of sulfur metabolism during pseudohyphal differentiation in *S. cerevisiae*

Our previous data showed that the perturbation of glycolysis using specific inhibitors, including 2DG/NaCi or deletion of genes that encode for glycolytic enzymes, including *pfk1/adh1* resulted in

a significant reduction in pseudohyphal differentiation in a cAMP-PKA-independent manner, under nitrogen-limiting conditions (*Figure 1*). Given that glycolysis strongly influences the global transcriptional landscape of *S. cerevisiae* (*Newcomb et al., 2003*; *Wu et al., 2019*), we undertook a comprehensive transcriptomic approach to identify the changes in the transcriptional landscape of *S. cerevisiae* colonies, in conditions that perturb glycolysis, resulting in the inhibition of pseudohyphal differentiation. We performed comparative RNA-Seq analysis using RNA isolated from wild-type cells spotted on SLAD in the presence and absence of the glycolysis inhibitor, 2DG (2DG addition was used to perturb glycolysis in our transcriptomics experiments, as it exerted the strongest inhibitory effect on pseudohyphal differentiation as shown in *Figure 1B and C*). The experimental schematic is depicted in *Figure 2A*. Briefly, comparative RNA-Seq analysis was done using wild-type colonies grown on SLAD in the presence and absence of 2DG isolated on day 5 (D-5) and day 10 (D-10), respectively (*Figure 2A*). We have used D-5 as an early time point for our experiments, as we have consistently observed the emergence of pseudohyphae in colonies spotted on SLAD at this time point and D-10 as a late time point wherein colonies exhibit robust pseudohyphal differentiation (*Figure 2—figure supplement 1A*). A volcano plot of differential gene expression in 2DG-treated cells compared to untreated cells in both D-5 and D-10 is shown in *Figure 2B* wherein some of the significantly upregulated and downregulated genes are highlighted with green colour. Previous studies in the field have shown that, under nitrogen-limiting/glucose-replete conditions, glycolytic intermediates provide various precursors needed for the biosynthesis of several amino acids in order to maintain amino acid homeostasis (*Dikicioglu et al., 2011*; *Martíez-Force and Benítez, 1992*) and perturbation of glycolysis under these conditions results in the increased transcription of various genes involved in amino acid biosynthesis, as a feedback response. As expected, the expression of several genes involved in amino acid biosynthesis and transport were significantly upregulated in 2DG-treated cells compared to untreated cells, in both D-5 and D-10, respectively (*Figure 2—figure supplement 1B*) corroborating the observations from the previously mentioned studies (*Dikicioglu et al., 2011*; *Martíez-Force and Benítez, 1992*; *Wu et al., 2004*). Interestingly and rather unexpectedly, the expression of multiple genes specifically involved in the biosynthesis and transport of sulfur-containing amino acids, cysteine and methionine were significantly downregulated in 2DG-treated cells compared to untreated cells in both D-5 and D-10, respectively (*Figure 2C*). Our RNA-Seq analysis clearly demonstrates that multiple genes involved in the de novo biosynthesis of sulfur-containing amino acids are significantly downregulated in the presence of 2DG, under nitrogen-limiting conditions.

Given that, perturbation of glycolysis via 2DG addition resulted in a clear downregulation of multiple genes involved in the de novo biosynthesis of sulfur-containing amino acids under nitrogen-limiting conditions, we wanted to corroborate these observations by using pertinent genetic knockout strains, ΔΔpfk1 and ΔΔadh1. In order to determine this, we spotted *S. cerevisiae* wild-type Σ1278b along with ΔΔpfk1 and ΔΔadh1 on SLAD and cells from colonies were isolated after ~4 days for RNA isolation. We then performed comparative RT-qPCR to check for the relative expression of genes involved in de novo biosynthesis of sulfur-containing amino acids, including *met32* (Zinc-finger DNA-binding transcription factor), *met3* (ATP sulfurylase), *met5* (Sulfite reductase beta subunit), and *met17* (O-acetyl homoserine-O-acetyl serine sulfhydrylase) (*Blaiseau et al., 1997*; *Masselot and Robichon-Szulmajster, 1975*). Our data clearly shows that all the aforesaid genes were significantly downregulated in the ΔΔpfk1 and ΔΔadh1 strains compared to wild-type (*Figure 2—figure supplement 1C*), suggesting that genetic perturbation of glycolysis negatively impacts sulfur metabolism under nitrogen-limiting conditions.

In order to determine whether these observed trends are reflected at the protein level, we generated strains wherein various proteins involved in the de novo biosynthesis of sulfur-containing amino acids, including Met4 (Leucine-zipper transcriptional activator), Met32 (Zinc-finger DNA-binding transcription factor), Met16 (3'-phosphoadenylsulfate reductase), Met10 (Subunit alpha of assimilatory sulfite reductase), Cys4 (Cystathionine beta-synthase), and Cys3 (Cystathionine gamma-lyase) were epitope-tagged (*Figure 2D*). These strains were spotted on SLAD in the presence and absence of 2DG, following which cells from colonies were isolated on D-5, and levels of these proteins were assessed by using Western blotting (*Figure 2E*). Our results indicate that the levels of Met4, Met32, Met16, Met10, Cys4, and Cys3 were significantly reduced in the 2DG-treated cells compared to untreated cells, corroborating our transcriptome data (*Figure 2E*, *Figure 2—figure supplement 2A*). This clearly demonstrates that perturbation of glycolysis negatively influences the expression

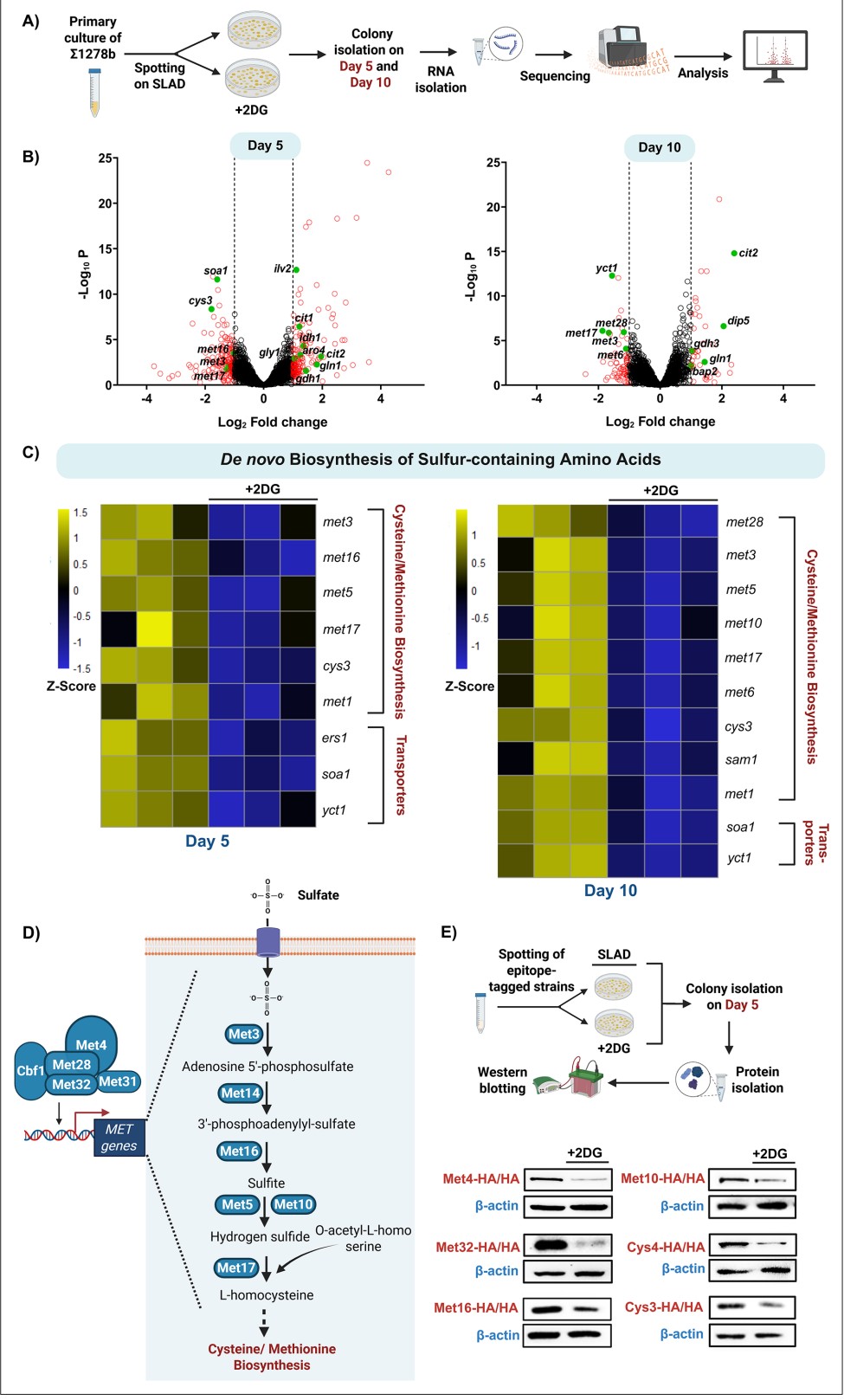

**Figure 2.** Comparative transcriptomics identifies the role of glycolysis in regulating sulfur metabolism during pseudohyphal differentiation in *S. cerevisiae*. (**A**) Schematic overview of steps involved in RNA isolation and sequencing. (**B**) Volcano plots represent differentially expressed genes in 2-Deoxy-D-Glucose (2DG)-treated cells compared to untreated cells isolated from D-5 and D-10 colonies, respectively. Genes which are upregulated

*Figure 2 continued on next page*

*Figure 2 continued*

(Log$_2$ fold change ≥ 1) and downregulated (Log$_2$ fold change ≤ –1) are highlighted in red. Some of the significantly upregulated and downregulated genes involved in the biosynthesis of various amino acids are highlighted in green. (**C**) Heatmaps represent the differentially expressed genes involved in de novo biosynthesis and transport of sulfur-containing amino acids, in 2DG-treated cells compared to untreated cells isolated from D-5 and D-10 colonies, respectively (n=3). Scale bar represents Z-Score. (**D**) Schematic overview of de novo biosynthesis of sulfur-containing amino acids. (**E**) Schematic overview of steps involved in the Western blotting experiments. Epitope-tagged strains of various proteins involved in the de novo biosynthesis of sulfur-containing amino acids, including Met4, Met32, Met16, Met10, Cys4, and Cys3 were spotted on SLAD in the presence and absence of sub-inhibitory concentration of 2DG, following which, cells from colonies were isolated on D-5 and levels of these proteins were assessed using Western blotting. β-actin was used as loading control. This figure was created using Biorender.com.

The online version of this article includes the following source data and figure supplement(s) for figure 2:

**Source data 1.** Uncropped and labeled blots for *Figure 2*.

**Source data 2.** Raw unedited blots for *Figure 2*.

**Figure supplement 1.** Comparative transcriptomics identifies the role of glycolysis during pseudohyphal differentiation in *S. cerevisiae*.

**Figure supplement 2.** Expression of proteins involved in the de novo biosynthesis of sulfur-containing amino acids in response to glycolysis perturbation.

**Figure supplement 2—source data 1.** Uncropped and labeled blots for *Figure 2—figure supplement 2*.

**Figure supplement 2—source data 2.** Raw unedited blots for *Figure 2—figure supplement 2*.

of sulfur metabolism proteins under nitrogen-limiting conditions. We next asked if the glycolysis-dependent regulation of sulfur metabolism under nitrogen-limiting conditions (SLAD) is restricted to conditions wherein cells physically attach to solid surfaces (colony growth) or whether it is conserved even in conditions where cells are free-floating (liquid-culture growth). In order to do this, pertinent epitope-tagged strains (wherein Met32, Met16, Met10, and Cys3 were HA-tagged) were grown in liquid SLAD in the presence and absence of 2DG for 24 hr after which the expression of the aforesaid proteins were assessed using Western blotting (*Figure 2—figure supplement 2B and C*). Importantly, our results indicate that the levels of Met32, Met16, Met10, and Cys3 were significantly reduced in the 2DG-treated cells compared to untreated cells (*Figure 2—figure supplement 2B and C*). Taken together, our data clearly demonstrate that the perturbation of glycolysis under nitrogen-limiting conditions negatively influences sulfur metabolism.

## Exogenous supplementation of sulfur sources rescues pseudohyphal differentiation defects caused by the perturbation of glycolysis in *S. cerevisiae*

Our comparative transcriptomics data clearly demonstrate that, under nitrogen-limiting conditions, glycolysis plays a critical role in regulating the expression of genes involved in the de novo biosynthesis of sulfur-containing amino acids (*Figure 2*). Based on our previous data, we hypothesized that under nitrogen-limiting conditions, active glycolysis enables de novo biosynthesis of sulfur-containing amino acids which, in turn, is critical for pseudohyphal differentiation and perturbation of active glycolysis using inhibitors like 2DG or disruption of genes involved in glycolysis, including *pfk1* or *adh1* negatively affects glycolysis-mediated regulation of sulfur metabolism leading to attenuation of pseudohyphal differentiation. If the proposed hypothesis holds true, the exogenous supplementation of sulfur-containing compounds should mitigate the fungal morphogenetic defects resulting from the perturbation of glycolysis. In order to test this, we performed sulfur add-back assays wherein *S. cerevisiae* wild-type Σ1278b was spotted on SLAD containing sub-inhibitory concentration of 2DG in the presence and absence of sulfur-containing amino acids (cysteine or methionine). Remarkably, the exogenous addition of cysteine resulted in a significant rescue of the pseudohyphal differentiation defect observed in nitrogen-limiting medium containing 2DG (*Figure 3A*). Corroborating this observation, the percentage of pseudohyphal cells in colonies isolated from SLAD containing 2DG and cysteine were significantly higher than the percentage of pseudohyphal cells in colonies isolated from SLAD containing 2DG alone (*Figure 3B*). Interestingly, the addition of methionine failed to rescue the pseudohyphal differentiation defects caused by the perturbation of glycolysis (*Figure 3A and B*).

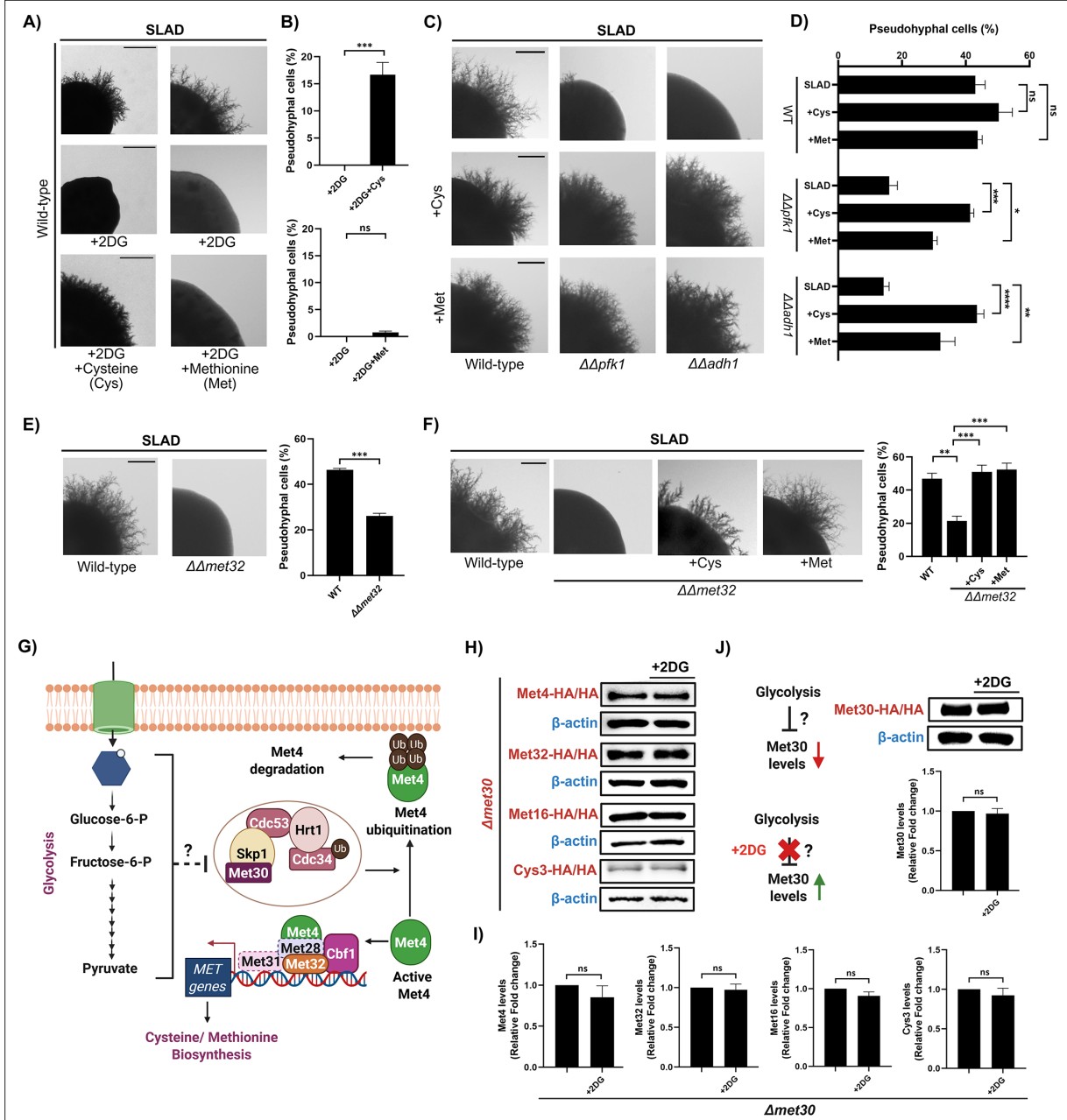

**Figure 3.** Glycolysis-mediated regulation of sulfur metabolism is critical for pseudohyphal differentiation in *S. cerevisiae*. (**A**) Wild-type Σ1278b was spotted on SLAD containing sub-inhibitory concentration of 2-Deoxy-D-Glucose (2DG) with and without sulfur-containing compounds, including cysteine (500 μM) or methionine (20 μM), incubated for 10 days at 30 °C. Scale bar represents 1 mm for whole colony images. (**B**) Cells from colonies were isolated and the length/width ratio of individual cells was measured using ImageJ, and the percentage of pseudohyphal cells from the total population was determined. More than 500 cells were counted for each condition. Statistical analysis was done using unpaired t-test, ***(p<0.001) and ns (non-significant). Error bars represent SEM. (**C**) Pertinent knockout strains (*ΔΔpfk1* or *ΔΔadh1*) along with wild-type Σ1278b were spotted on SLAD with and without sulfur-containing compounds, including cysteine (200 μM) or methionine (20 μM), incubated for 10 days at 30 °C. Scale bar represents 1 mm for whole colony images. (**D**) Cells from colonies were isolated and the length/width ratio of individual cells was measured using ImageJ, and the percentage of pseudohyphal cells from the total population was determined. More than 500 cells were counted for each condition. Statistical analysis was done using one-way ANOVA test, ****(p<0.0001), ***(p<0.001), **(p<0.01), *(p<0.05) and ns (non-significant). Error bars represent SEM. (**E**) Pertinent knockout strain which lack genes encoding for a transcription factor involved in de novo biosynthesis of sulfur-containing amino acids, including *met32* along with wild-type Σ1278b were spotted on SLAD. Scale bar represents 1 mm for whole colony images. Cells from colonies were isolated, and the length/width ratio of individual cells was measured using ImageJ, and the percentage of pseudohyphal cells from the total population was determined. More than 500 cells were counted for each condition. Statistical analysis was done using unpaired t-test, ***(p<0.001). Error bars represent SEM. (**F**) Pertinent knockout strain (*ΔΔmet32)* was spotted on SLAD with and without sulfur-containing compounds, including cysteine (200 μM) or methionine

*Figure 3 continued on next page*

*Figure 3 continued*

(20 μM), incubated for 10 days at 30 °C. Wild-type spotted on SLAD was used as control. Scale bar represents 1 mm for whole colony images. Cells from colonies were isolated, and the length/width ratio of individual cells was measured using ImageJ and the percentage of pseudohyphal cells from the total population was determined. More than 500 cells were counted for each condition. Statistical analysis was done using one-way ANOVA test, ***($p<0.001$) and **($p<0.01$). Error bars represent SEM. (**G**) Schematic overview of the proposed role of SCF^Met30 in glycolysis-dependent regulation of sulfur metabolism during pseudohyphal differentiation under nitrogen-limiting conditions in *S. cerevisiae*. (**H**) Epitope-tagged strains of various proteins involved in the de novo biosynthesis of sulfur-containing amino acids, including Met4, Met32, Met16, and Cys3 in the *Δmet30* background were spotted on SLAD in the presence and absence of sub-inhibitory concentration of 2DG, following which, cells from colonies were isolated on D-5 and levels of these proteins were assessed using Western blotting. β-actin was used as loading control. (**I**) Raw images of Western blots were analyzed using ImageJ to normalize targeted protein expression with the expression of housekeeping protein- β-actin, in order to generate densitometric graphs. Statistical analysis was done using unpaired t-test, ns (non-significant). Error bars represent SEM. (**J**) Epitope-tagged strain of Met30 was spotted on SLAD in the presence and absence of sub-inhibitory concentration of 2DG following which cells from colonies were isolated on D-5 and levels of these proteins were assessed using Western blotting. β-actin was used as loading control. Raw images of Western blots were analyzed using ImageJ to normalize targeted protein expression with the expression of housekeeping protein-β-actin, in order to generate densitometric graphs. Statistical analysis was done using unpaired t-test, ns (non-significant). Error bars represent SEM. This figure was created using Biorender.com.

The online version of this article includes the following source data and figure supplement(s) for figure 3:

**Source data 1.** Uncropped and labeled blots for *Figure 3*.

**Source data 2.** Raw unedited blots for *Figure 3*.

**Figure supplement 1.** Monitoring of overall growth rate of *ΔΔmet32* and *Δmet30* strains.

We next asked if the exogenous addition of sulfur sources rescues the pseudohyphal differentiation defect exhibited by glycolysis-defective strains, *ΔΔpfk1* and *ΔΔadh1*. In order to do this, we performed similar sulfur add-back assays wherein *S. cerevisiae* wild-type Σ1278b along with *ΔΔpfk1* and *ΔΔadh1* strains were spotted on SLAD in the presence and absence of sulfur-containing amino acids (cysteine or methionine). The exogenous addition of cysteine and methionine were able to significantly rescue the pseudohyphal differentiation defect exhibited by the *ΔΔpfk1* and *ΔΔadh1* strains albeit at different levels (*Figure 3C*). Corroborating this observation, the percentage of pseudohyphal cells in *ΔΔpfk1* and *ΔΔadh1* colonies isolated from SLAD containing cysteine or methionine were significantly higher than the percentage of pseudohyphal cells in *ΔΔpfk1* and *ΔΔadh1* colonies isolated from SLAD alone (*Figure 3D*). Taken together, our results clearly demonstrate that the glycolysis-mediated regulation of sulfur metabolism is critical for pseudohyphal differentiation under nitrogen-limiting conditions.

## Sulfur metabolism is critical for pseudohyphal differentiation under nitrogen-limiting conditions in *S. cerevisiae*

Our previous observations strongly suggest a direct role for sulfur metabolism in regulating fungal morphogenesis under nitrogen-limiting conditions. In order to determine whether glycolysis-dependent sulfur metabolism is critical for pseudohyphal differentiation under nitrogen-limiting conditions, we generated a knockout strain that lacks *met32* (gene that encodes for a zinc-finger DNA-binding transcription factor involved in the regulation of de novo biosynthesis of sulfur-containing amino acids *Blaiseau et al., 1997*). Unlike deletions of most other genes involved in the de novo biosynthesis of sulfur-containing amino acids, the *met32* deletion strain does not exhibit auxotrophy for methionine or cysteine (*Blaiseau et al., 1997*). This is likely due to the functional redundancy provided by its paralog, *met31* (*Blaiseau et al., 1997*). We then assessed the ability of the *met32* deletion strain to undergo pseudohyphal differentiation under nitrogen-limiting conditions. Deletion of *met32* resulted in a significant reduction in pseudohyphal differentiation. Corroborating this observation, the percentage of pseudohyphal cells in *ΔΔmet32* colonies were significantly lower in SLAD, compared to *S. cerevisiae* wild-type Σ1278b colonies (*Figure 3E*). This suggests that perturbations to sulfur metabolism negatively affect pseudohyphal differentiation. In order to rule out whether the reduction in pseudohyphal differentiation in this mutant is due to a general growth defect, we performed growth curve analysis wherein overnight grown cultures of deletion strain (*ΔΔmet32*) along with wild-type strain (Σ1278b) were diluted to $OD_{600}=0.01$ in fresh SLAD and allowed to grow at 30 °C for 24 hr. $OD_{600}$ was recorded at 3 hr intervals (*Figure 3—figure supplement 1A*). Our data suggests that the reduction in pseudohyphal differentiation is not due to general growth defect, as the mutant was able to grow similar to the wild-type strain in SLAD. We next asked if the exogenous addition of sulfur sources rescues the pseudohyphal differentiation defect exhibited by *ΔΔmet32*. In order to do

this, we performed a simple sulfur add-back assay, wherein the $\Delta\Delta met32$ strain was spotted on SLAD containing either cysteine or methionine and allowed to undergo pseudohyphal differentiation. Our results show that the addition of cysteine or methionine completely rescued the pseudohyphal differentiation defect exhibited by $\Delta\Delta met32$ strain. The percentage of pseudohyphal cells were also significantly higher in $\Delta\Delta met32$ colonies growing on SLAD containing cysteine or methionine compared to just SLAD (*Figure 3F*). Taken together, our data clearly demonstrate that the efficient de novo biosynthesis of sulfur-containing amino acids, including cysteine and methionine is critical for pseudohyphal differentiation under nitrogen-limiting conditions.

## Met30 is involved in the glycolysis-dependent regulation of sulfur metabolism during pseudohyphal differentiation in *S. cerevisiae*

Our previous data clearly demonstrate that, under nitrogen-limiting conditions, the levels of Met4, a key activator of genes involved in the de novo biosynthesis of sulfur-containing amino acids, was significantly reduced in 2DG-treated cells compared to untreated cells (*Figure 2E*). Met4 activity is known to be negatively regulated by the SCF$^{Met30}$ (Skp1/Cullin1/F-box) ubiquitin ligase complex in *S. cerevisiae* (*Rouillon et al., 2000*). Under sulfur-replete conditions, Met30 promotes Met4 ubiquitination and degradation, thereby repressing the *MET* gene network. Conversely, sulfur limitation reduces Met30 activity, allowing Met4 to become active and induce the expression of genes required for sulfur amino acid biosynthesis (*Smothers et al., 2000*). It is important to note that deletion of *met30* is lethal in haploid strains of *S. cerevisiae* (*Kaiser et al., 1998*). While point mutations or temperature-sensitive *met30* mutants are commonly used in haploid strains to circumvent this lethality (*Kaiser et al., 1998*; *Su et al., 2008*), our study in diploid *S. cerevisiae* involved generating a heterozygous viable *met30* ($\Delta met30$) deletion strain. In order to confirm that the deletion of a single copy of *met30* does not lead to any growth defects in SLAD, we performed growth curve analysis wherein overnight grown cultures of deletion strain ($\Delta met30$) along with wild-type strain ($\Sigma1278b$) were diluted to $OD_{600}=0.01$ in fresh SLAD and allowed to grow at 30 °C for 24 hr. $OD_{600}$ was recorded at 3 hr intervals (*Figure 3—figure supplement 1A*). Our data suggest that the deletion of *met30* does not affect overall growth, as the mutant was able to grow similar to the wild-type strain in SLAD (*Figure 3—figure supplement 1A*). Based on our previous data, we hypothesized that under nitrogen-limiting conditions, active glycolysis enables de novo biosynthesis of sulfur-containing amino acids by reducing the activity of Met30. Conversely, we hypothesized that perturbing active glycolysis with the glycolysis inhibitor 2DG increases Met30 activity, which in turn inactivates Met4, thereby negatively affecting *MET* gene transcription, ultimately resulting in attenuated pseudohyphal differentiation (*Figure 3G*). In order to investigate this hypothesis, we generated strains wherein one copy of *met30* was deleted and various proteins involved in the de novo biosynthesis of sulfur-containing amino acids, including Met4 (Leucine-zipper transcriptional activator), Met32 (Zinc-finger DNA-binding transcription factor), Met16 (3'-phosphoadenylsulfate reductase), and Cys3 (Cystathionine gamma-lyase) were epitope-tagged. These strains were spotted on SLAD in the presence and absence of 2DG, following which cells from colonies were isolated on D-5 and levels of these proteins were assessed by using Western blotting. Our results indicate that the levels of Met4, Met32, Met16, and Cys3 were similar in both 2DG-treated and untreated cells, in the $\Delta met30$ deletion strain (*Figure 3H and I*). This suggests that glycolysis influences Met30 activity, which in turn regulates Met4 activity, thereby affecting the expression of genes involved in the de novo biosynthesis of sulfur-containing amino acids under nitrogen-limiting conditions.

Based on our previous observation that 2DG treatment significantly reduces Met4 levels in nitrogen-limiting conditions (an effect that was notably absent in *met30* deletion strains), we hypothesized that the 2DG-induced downregulation of Met4 is mediated by an increase in Met30 levels. In order to check Met30 levels in the presence of 2DG, we generated a strain wherein Met30 was epitope-tagged, and this strain was spotted on SLAD, in the presence and absence of 2DG, following which cells from colonies were isolated on D-5 and levels of Met30 were assessed by using Western blotting. Our results indicate that the levels of Met30 were similar in both 2DG-treated and untreated cells (*Figure 3J*), suggesting that glycolysis influences Met30 activity post-translationally rather than through changes in its protein abundance. Collectively, our findings indicate that the perturbation of glycolysis under nitrogen-limiting conditions negatively impacts sulfur metabolism through a mechanism dependent on Met30 activity.

## Glycolysis-dependent sulfur metabolism is critical for hyphal differentiation in *C. albicans*

*C. albicans* is an opportunistic fungal pathogen that exhibits a remarkable degree of fungal morphogenesis in response to various environmental cues (*Huang, 2012*). Yeast to hyphal differentiation in *C. albicans* is well studied. However, we do not have a complete understanding of fundamental metabolic drivers that orchestrate this phenomenon. Given that central carbon metabolic pathways, including glycolysis, are well conserved across a plethora of fungal species, including *C. albicans* (*Askew et al., 2009*) and nitrogen limitation is one of the key triggers for yeast to hyphal differentiation in *C. albicans* (*Csank et al., 1998*), we wanted to determine if glycolysis-mediated sulfur metabolism plays an important role in this morphogenetic switching in *C. albicans* as well. First, we wanted to understand whether the ability of C. *albicans* cells to metabolize glucose under nitrogen-limiting conditions is critical for fungal morphogenesis (*Figure 4A*). In order to do this, *C. albicans* wild-type SC5314 was spotted on SLAD, in the presence and absence of sub-inhibitory concentration of glycolysis inhibitor, 2DG, allowed to grow for 7 days, and cells were isolated and imaged using bright-field microscopy. Our data clearly demonstrate that sub-inhibitory concentration of 2DG significantly reduced filamentation in *C. albicans* (*Figure 4B*) without compromising the overall growth of *C. albicans* wild-type SC5314 (*Figure 4H*). We then isolated cells from these colonies and quantified the percentage of hyphal cells (cells with a length/width ratio of 4.5 or more *Su et al., 2018*) in these colonies. Corroborating our microscopy data, the percentage of hyphal cells in colonies spotted on SLAD containing 2DG was significantly lower compared to the percentage of hyphal cells in colonies spotted on SLAD without 2DG (*Figure 4B*). To investigate whether the hyphal differentiation defects that we observed due to glycolysis perturbation were solely due to the inability of glucose to activate the cAMP-PKA pathway, we performed cAMP add-back assays wherein *C. albicans* wild-type SC5314 was spotted on SLAD containing sub-inhibitory concentrations of 2DG in the presence and absence of 5 mM cAMP. Interestingly, corroborating our *S. cerevisiae* data, the exogenous addition of cAMP failed to rescue hyphal differentiation defect caused by the perturbation of glycolysis through 2DG under nitrogen-limiting conditions in *C. albicans* (*Figure 4—figure supplement 1B*). Taken together, our results show that the ability of *C. albicans* to metabolize glucose is critical for fungal morphogenesis in a cAMP-PKA-independent manner under nitrogen-limiting conditions.

Given that the perturbation of glycolysis under nitrogen-limiting conditions, using 2DG exhibited a strong inhibitory effect on hyphal differentiation in *C. albicans* (*Figure 4B*), similar to what we observed in *S. cerevisiae*, we wanted to determine if the expression of genes involved in the de novo biosynthesis of sulfur-containing amino acids are affected when glycolysis is perturbed in *C. albicans*, under nitrogen-limiting conditions (similar to what we observed in *S. cerevisiae*). In order to determine this, we spotted *C. albicans* wild-type SC5314 on SLAD, in the presence and absence of 2DG, and cells from colonies were isolated after ~4 days for RNA isolation. We then performed comparative RT-qPCR to check for the relative expression of genes involved in de novo biosynthesis of sulfur-containing amino acids, including *met32* (DNA-binding transcription factor), *met3* (ATP sulfurylase), *met5* (*ecm17*) (Sulfite reductase beta subunit), *met10* (Sulfite reductase), and *met17* (*met15*) (O-acetyl homoserine-O-acetyl serine sulfhydrylase) (*Chebaro et al., 2017*; *Li et al., 2013*; *Shrivastava et al., 2021*), in cells isolated from colonies grown on SLAD containing 2DG compared to cells isolated from colonies grown on SLAD without 2DG (*Figure 4C*). Our data clearly shows that all the aforesaid genes involved in de novo biosynthesis of sulfur-containing amino acids pathway, including *met32*, *met3*, *met5* (*ecm17*), *met10,* and *met17* (*met15*) were significantly downregulated in the presence of 2DG (*Figure 4D*) suggesting that perturbation of glycolysis negatively impacts sulfur metabolism in *C. albicans* under nitrogen-limiting conditions. To determine if the significant reduction in hyphal differentiation that we observed in the presence of 2DG can be attributed to the reduced expression of genes involved in the de novo biosynthesis of sulfur-containing amino acids, we performed sulfur add-back assays similar to what was done in *S. cerevisiae* (*Figure 3A*), wherein *C. albicans* wild-type SC5314 was spotted on SLAD containing 2DG in the presence and absence of cysteine or methionine, allowed to grow for 7 days, and cells were isolated and imaged using bright-field microscopy. The addition of cysteine or methionine significantly rescued hyphal differentiation defects in the presence of 2DG (*Figure 4E and F*). We then isolated cells from these colonies and quantified the percentage of hyphal cells. Corroborating our microscopy data, the percentage of hyphal cells in colonies spotted on SLAD

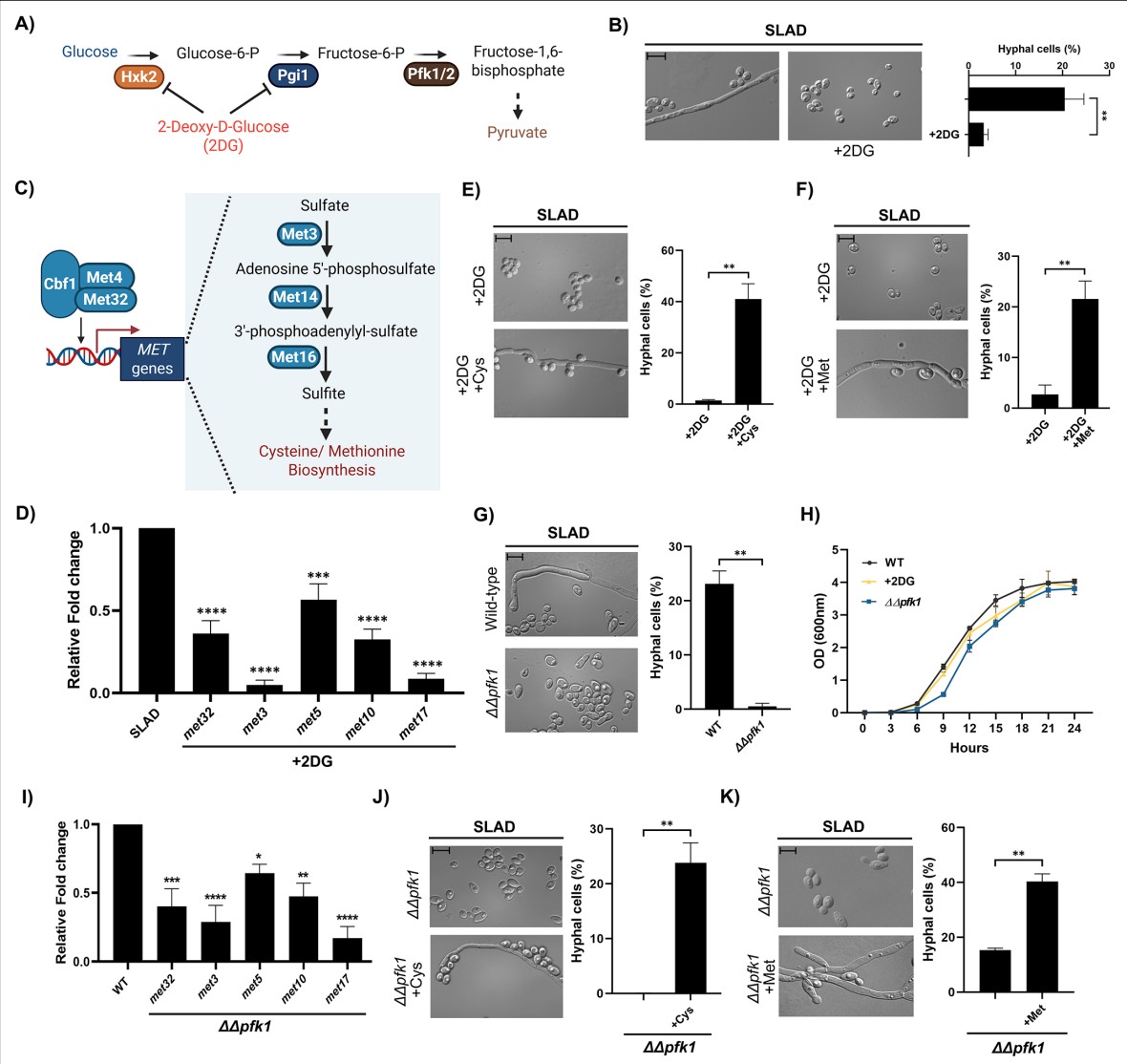

**Figure 4.** Glycolysis-dependent sulfur metabolism is critical for hyphal differentiation in *C. albicans*. (**A**) Schematic overview of glycolysis and its inhibition by glycolysis inhibitor 2-Deoxy-D-Glucose (2DG). (**B**) Wild-type SC5314 was spotted on SLAD with and without sub-inhibitory concentration of glycolysis inhibitor 2DG, incubated for 7 days at 37 °C. Cells from colonies were isolated and the length/width ratio of individual cells was measured using ImageJ and the percentage of hyphal cells from the total population was determined. More than 500 cells were counted for each condition. Scale bar represents 10 μm for single cell images. Statistical analysis was done using unpaired t-test, **(p<0.01). Error bars represent SEM. (**C**) Schematic overview of de novo biosynthesis of sulfur-containing amino acids. (**D**) Wild-type SC5314 was spotted on SLAD, in the presence and absence of sub-inhibitory concentration of 2DG and cells from colonies were isolated after ~4 days, for RNA isolation. We then performed comparative RT-qPCR to check for the relative expression of genes (each gene was normalized to its respective control group) involved in the de novo biosynthesis of sulfur-containing amino acids pathway, including *met32*, *met3*, *met5* (*ecm17*), *met10,* and *met17* (*met15*). Statistical analysis was done using one-way ANOVA test, ****(p<0.0001) and ***(p<0.001). Error bars represent SEM. (**E**) Wild-type SC5314 was spotted on SLAD containing sub-inhibitory concentration of 2DG with and without cysteine (100 μM) and incubated for 7 days at 37 °C. Cells from colonies were isolated and the length/width ratio of individual cells was measured using ImageJ and the percentage of hyphal cells from the total population was determined. Scale bar represents 10 μm for single cell images. More than 500 cells were counted for each condition. Statistical analysis was done using unpaired t-test, **(p<0.01). Error bars represent SEM. (**F**) Wild-type SC5314 was spotted on SLAD containing sub-inhibitory concentration of 2DG with and without methionine (20 μM) and incubated for 7 days at 37 °C. Cells from colonies were isolated and the length/width ratio of individual cells was measured using ImageJ and the percentage of hyphal cells from the total population was determined. More than 500 cells were counted for each condition. Scale bar represents 10 μm for single cell images. Statistical analysis was done using unpaired t-test, **(p<0.01). Error bars represent SEM. (**G**) Wild-type SC5314 and pertinent knockout strain which lacks the genes encoding for a glycolytic enzyme, *pfk1* (*ΔΔpfk1*) were spotted on SLAD, incubated for 7 days at 37 °C. Cells from colonies were isolated and the length/width ratio of individual cells was measured using ImageJ and the percentage of hyphal cells from the total population was determined. Scale bar represents 10 μm for single cell images. More than 500 cells were counted for each strain. Statistical analysis was done

*Figure 4 continued on next page*

*Figure 4 continued*

using unpaired t-test, **(p<0.01). Error bars represent SEM. (**H**) Growth curve was performed to monitor overall growth of wild-type strain on SLAD and SLAD+2DG and ΔΔpfk1 on SLAD. Overnight grown culture of wild-type strain (SC5314) was diluted to $OD_{600}$=0.01 in fresh SLAD medium with and without 2DG (0.2% (w/v)) and allowed to grow at 30 °C for 24 hr. $OD_{600}$ was recorded at 3 h intervals. For ΔΔpfk1, overnight grown culture of ΔΔpfk1 strains along with wild-type strain (SC5314) were diluted to $OD_{600}$=0.01 in fresh SLAD and allowed to grow at 30 °C for 24 hr. $OD_{600}$ was recorded at 3 hr intervals. (**I**) ΔΔpfk1 along with Wild-type SC5314 was spotted on SLAD and cells from colonies were isolated after ~4 days, for RNA isolation. We then performed comparative RT-qPCR to check for the relative expression of genes (each gene was normalized to its respective control group) involved in the de novo biosynthesis of sulfur-containing amino acids pathway, including *met32*, *met3*, *met5* (*ecm17*), *met10*, and *met17* (*met15*). Statistical analysis was done using one-way ANOVA test, ****(p<0.0001), ***(p<0.001), **(p<0.01), and *(p<0.05). Error bars represent SEM. (**J**) ΔΔpfk1 was spotted on SLAD in the presence and absence of cysteine (100 µM), incubated for 7 days at 37 °C. Cells from colonies were isolated and the length/width ratio of individual cells was measured using ImageJ and the percentage of hyphal cells from the total population was determined. Scale bar represents 10 µm for single cell images. More than 500 cells were counted for each condition. Statistical analysis was done using unpaired t-test, **(p<0.01). Error bars represent SEM. (**K**) ΔΔpfk1 was spotted on SLAD in the presence and absence of methionine (10 µM), incubated for 7 days at 37 °C. Cells from colonies were isolated and the length/width ratio of individual cells was measured using ImageJ and the percentage of hyphal cells from the total population was determined. Scale bar represents 10 µm for single cell images. More than 500 cells were counted for each condition. Statistical analysis was done using unpaired t-test, **(p<0.01). Error bars represent SEM. This figure was created using Biorender.com.

The online version of this article includes the following figure supplement(s) for figure 4:

**Figure supplement 1.** Confirmation of *pfk1* deletion in *C. albicans* and glycolysis is critical for hyphal differentiation in a cAMP-PKA-independent manner in *C. albicans*.

---

containing 2DG with cysteine or methionine were significantly higher compared to the percentage of hyphal cells in colonies spotted on SLAD with 2DG alone (***Figure 4E and F***).

Our previous results clearly demonstrate that glycolysis and glycolysis-dependent sulfur metabolism is critical for fungal morphogenesis of *C. albicans* under nitrogen-limiting conditions. In order to corroborate these observations by using pertinent genetic knockout strains that are compromised in performing glycolysis, we generated a double knockout strain lacking the gene that encodes for the glycolytic enzyme, Pfk1 using the SAT-Flipper method (***Reuss et al., 2004***). The deletion of *pfk1* was confirmed through whole genome sequencing (***Figure 4—figure supplement 1A***). We then tested the ability of this deletion strain to undergo hyphal differentiation under nitrogen-limiting conditions. Briefly, *C. albicans* wild-type SC5314 and ΔΔpfk1 were spotted on SLAD, allowed to grow for 7 days, and cells were isolated and imaged using bright-field microscopy. Our result shows that the ΔΔpfk1 strain exhibits a significant reduction in hyphal differentiation compared to *C. albicans* wild-type SC5314, under nitrogen-limiting conditions (***Figure 4G***). We then isolated cells from these colonies and quantified the percentage of hyphal cells in these colonies. Corroborating our microscopy data, the percentage of hyphal cells in *C. albicans* wild-type SC5314 colonies spotted on SLAD were significantly higher compared to the percentage of hyphal cells in ΔΔpfk1 colonies (***Figure 4G***). In order to rule out whether the reduction in hyphal differentiation in this mutant is due to a general growth defect, we performed growth curve analysis wherein overnight grown cultures of deletion strain (ΔΔpfk1) along with wild-type strain (SC5314) were diluted to $OD_{600}$=0.01 in fresh SLAD and allowed to grow at 30 °C for 24 hr. $OD_{600}$ was recorded at 3 hr intervals. Our data suggest that the reduction in hyphal differentiation is not due to general growth defect, as the mutant was able to grow similar to the wild-type strain in SLAD (***Figure 4H***). We next asked if the exogenous addition of cAMP rescues the hyphal differentiation defect exhibited by the glycolysis-defective strain, ΔΔpfk1. In order to do this, we performed similar cAMP add-back assays wherein *C. albicans* wild-type SC5314 along with ΔΔpfk1 strain were spotted on SLAD in the presence and absence of 5 mM cAMP. Corroborating our previous data, the exogenous addition of cAMP failed to rescue hyphal differentiation defect caused by the perturbation of glycolysis via the deletion of *pfk1* (***Figure 4—figure supplement 1C***).

In order to determine if the expression of genes involved in the de novo biosynthesis of sulfur-containing amino acids are affected in ΔΔpfk1 under nitrogen-limiting conditions (similar to what we observed in the presence of 2DG, ***Figure 4D***), we spotted *C. albicans* wild-type SC5314 and ΔΔpfk1 on SLAD and cells from colonies were isolated after ~4 days for RNA isolation. We then performed comparative RT-qPCR to check for the relative expression of genes involved in de novo biosynthesis of sulfur-containing amino acids, including *met32* (DNA-binding transcription factor), *met3* (ATP sulfurylase), *met5* (*ecm17*) (Sulfite reductase beta subunit), *met10* (Sulfite reductase), and *met17* (*met15*) (O-acetyl homoserine-O-acetyl serine sulfhydrylase) (***Chebaro et al., 2017***; ***Li et al., 2013***; ***Shrivastava et al., 2021***; ***Figure 4C***). Our data clearly shows that all the aforesaid genes involved in de novo

biosynthesis of sulfur-containing amino acids pathway, including *met32*, *met3*, *met5* (*ecm17*), *met10*, and *met17* (*met15*) were significantly downregulated in *ΔΔpfk1* compared to the wild-type (*Figure 4I*) suggesting that the genetic perturbation of glycolysis negatively impacts sulfur metabolism in *C. albicans* under nitrogen-limiting conditions.

Our RT-qPCR results showed that the deletion of *pfk1* negatively influences sulfur metabolism, and this, in turn, might be causal for the reduced hyphal differentiation observed in these conditions. Given this, we wanted to determine if the hyphal differentiation defect exhibited by *ΔΔpfk1* can be rescued by the addition of sulfur-containing amino acids. In order to do this, we performed sulfur add-back assays wherein the *ΔΔpfk1* strain was spotted on SLAD in the presence and absence of cysteine or methionine, allowed to grow for 7 days and imaged using bright field microscopy. Interestingly, our data shows that cysteine or methionine were able to rescue the hyphal differentiation defect exhibited by the *ΔΔpfk1* strain (*Figure 4J and K*). We then isolated cells from these colonies and quantified the percentage of hyphal cells. Corroborating our microscopy data, the percentage of hyphal cells in *ΔΔpfk1* colonies spotted on SLAD containing cysteine or methionine were significantly higher compared to the percentage of hyphal cells in *ΔΔpfk1* colonies spotted on SLAD alone (*Figure 4J and K*). Overall, our data suggests that glycolysis-mediated regulation of sulfur metabolism is critical for hyphal differentiation of *C. albicans* under nitrogen-limiting conditions, highlighting the remarkable similarities with respect to the regulation of fungal morphogenesis between *S. cerevisiae* and *C. albicans*.

## Perturbation of glycolysis leads to attenuated fungal virulence in *C. albicans* which is rescued by sulfur supplementation

Several studies have established the importance of fungal morphogenesis as a key virulence strategy used by *C. albicans* to establish successful infections in the host (*Gow et al., 2011*; *Moyes et al., 2016*; *Wilson et al., 2016*). Based on our previous observations that perturbation of glycolysis, either through 2DG treatment or deletion of *pfk1* (*ΔΔpfk1*), significantly attenuates hyphal differentiation, we hypothesized that this observed phenotype could be a consequence of transcriptional downregulation of genes associated with hyphal differentiation and virulence. In order to test this hypothesis, we spotted *C. albicans* wild-type SC5314 on SLAD, in the presence and absence of 2DG or wild-type along with *ΔΔpfk1* on SLAD. Cells from colonies were isolated after ~4 days for RNA isolation. We then performed comparative RT-qPCR to check for the relative expression of genes involved in hyphal differentiation and virulence, including *als3* (Cell wall adhesin), *ece1* (Candidalysin), *hwp1* (Hyphal cell wall protein), *hyr1* (GPI-anchored hyphal cell wall protein), *ihd1* (GPI-anchored protein), *rbt1* (Cell wall protein), and *sap6* (Secreted aspartyl protease) (*Bailey et al., 1996*; *Deorukhkar and Roushani, 2017*; *Liu and Filler, 2011*; *Martin et al., 2013*; *Monniot et al., 2013*; *Moyes et al., 2016*; *Richardson et al., 2018*; *Sharkey et al., 1999*; *Figure 5A*). Our data clearly shows that all the aforesaid genes involved in hyphal differentiation and virulence, including *als3*, *ece1*, *hwp1*, *hyr1*, *ihd1*, *rbt1*, and *sap6* were significantly downregulated in both 2DG-treated cells and in *ΔΔpfk1*, suggesting that perturbation of glycolysis negatively affects the expression of genes involved in hyphal differentiation and virulence in *C. albicans* under nitrogen-limiting conditions (*Figure 5B*).

Given that *ΔΔpfk1* displayed a significant attenuation in its ability to undergo hyphal differentiation and it had significantly reduced expression of genes associated with hyphal differentiation and virulence, we wanted to determine if this deletion strain was compromised in its ability to infect the host. In order to do this, we took a two-pronged approach wherein we first tested the ability of the *ΔΔpfk1* strain to survive the harsh intracellular environment of the macrophages which *C. albicans* encounters during various stages of host infection (*Jiménez-López and Lorenz, 2013*; *May and Casadevall, 2018*; *Uwamahoro et al., 2014*). We performed an in vitro microbial survival assay wherein *RAW 264.7* macrophages were incubated either with *C. albicans* wild-type SC5314 or the *ΔΔpfk1* strain for 1 hr, following which macrophages were lysed and fungi were serially diluted and plated to determine the viable population of the *C. albicans* wild-type SC5314 and the *ΔΔpfk1* strain (*Figure 5C*). We also calculated the percentage of the wild-type or mutant strain that survived the incubation with the macrophages. Our results demonstrate that the viable population of *ΔΔpfk1* strain was significantly lower compared to the wild-type strain, after incubation with the macrophages (*Figure 5C*). Similarly, the percentage survival of the *ΔΔpfk1* strain was significantly lower than the wild-type SC5314

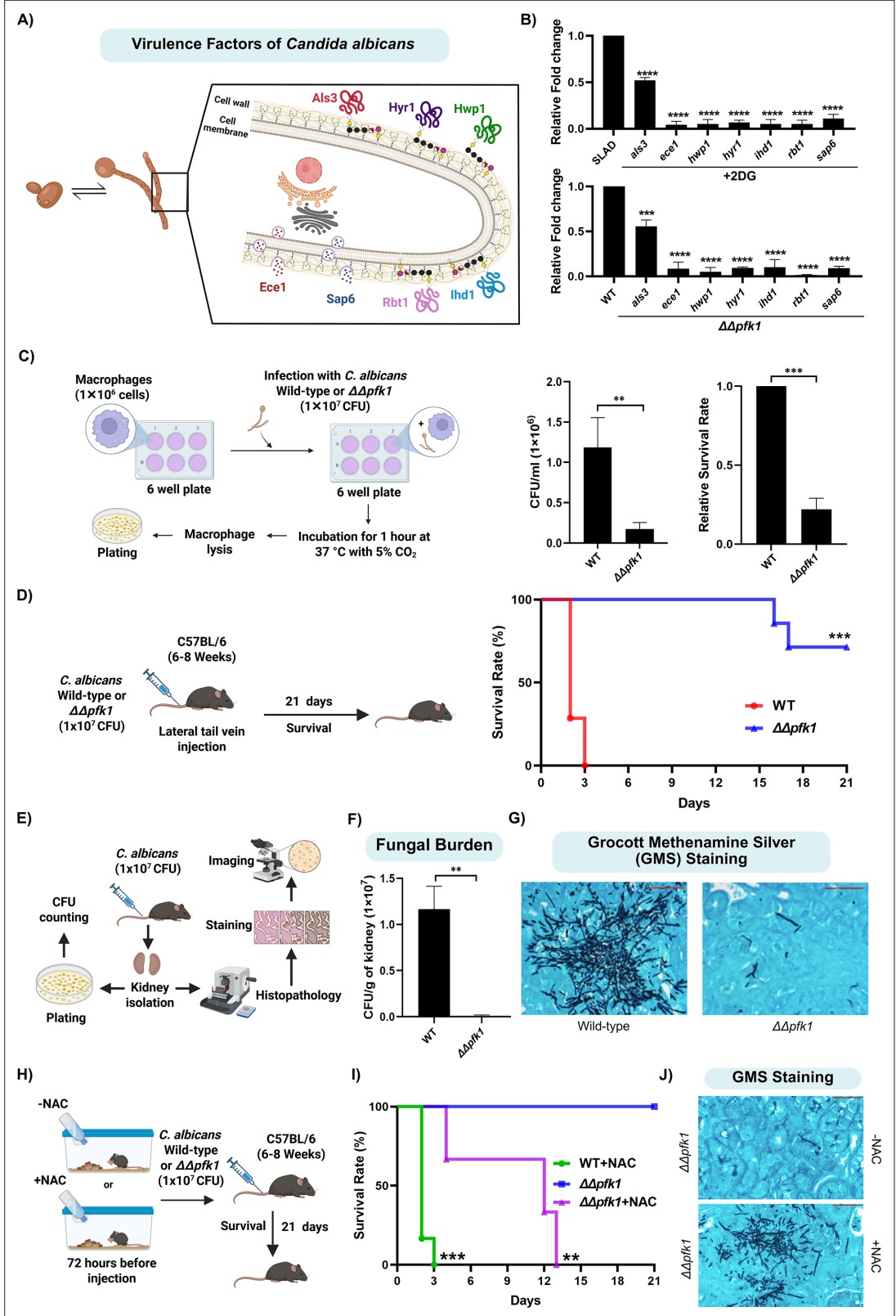

**Figure 5.** Perturbation of glycolysis leads to attenuated fungal virulence in *C. albicans* which is rescued by sulfur supplementation. (**A**) Schematic overview of various proteins involved in hyphal differentiation and virulence of *C. albicans*. (**B**) Wild-type SC5314 was spotted on SLAD, in the presence and absence of sub-inhibitory concentration of 2-Deoxy-D-Glucose (2DG) or wild-type SC5314 and *ΔΔpfk1* were spotted on SLAD and cells from these colonies were isolated after ~4 days, for RNA isolation. We then performed comparative RT-qPCR to check for the relative expression of genes (each

*Figure 5 continued on next page*

*Figure 5 continued*

gene was normalized to its respective control group) involved in hyphal differentiation and virulence of *C. albicans*, including *als3, ece1, hwp1, hyr1, ihd1, rbt1,* and *sap6*. Statistical analysis was done using one-way ANOVA test, ****($p<0.0001$) and ***($p<0.001$). Error bars represent SEM. (**C**) Schematic overview of in vitro microbial survival assay. *RAW 264.7* macrophages were incubated with the wild-type and *ΔΔpfk1* strain with MOI = 10. After 1 hr of incubation, macrophages were lysed and plating was done to enumerate colony-forming units (CFU). The percentage of survival was expressed as the number of CFU in the presence of macrophages divided by the number of CFU in the absence of macrophages. Statistical analysis was done using unpaired t-test, ***($p<0.001$) and **($p<0.01$). Error bars represent SEM. (**D**) 6–8 week-old C57BL/6 mice were infected intravenously via lateral tail vein injection with $1\times10^7$ CFU of wild-type SC5314 or $1\times10^7$ CFU of *ΔΔpfk1*. To determine the survival rate, mice were monitored for 21 days post-infection for clinical signs of illness or mortality. The results are representative of seven mice per injected strain. Survival percentage were statistically evaluated by the Log-rank Mantel-Cox test, ***($p<0.001$) (n=7). (**E**) Schematic overview of kidney isolation to check fungal burden using CFU counting and histology. (**F**) Fungal burden was measured by CFU counting after kidney isolation from mice injected with either wild-type or *ΔΔpfk1*. Statistical analysis was done using unpaired t-test, **($p<0.01$). Error bars represent SEM. (**G**) Histopathological analysis of kidney sections using Grocott Methanamine Silver (GMS) staining was done for kidneys isolated from mice injected with either wild-type SC5314 or *ΔΔpfk1*. Fungal cells appear black in colour against the colored background of the kidney tissue. Bright field imaging was done using a Zeiss Axioplan 2 microscope at 40X magnification. Scale bar represents 50 µm. (**H**) Schematic overview of N-acetyl cysteine (NAC) administration to mice. (**I**) 6 mg/ml of NAC was dissolved in distilled water and administered orally to 6–8 week-old C57BL/6 mice before 72 hr of *C. albicans* infection. Mice administered with normal distilled water were used as controls. After 72 hr of treatment, mice were infected intravenously via lateral tail vein injection with $1\times10^7$ CFU of SC5314 or $1\times10^7$ CFU of *ΔΔpfk1*. To determine the survival rate, mice were monitored for 21 days post-infection for clinical signs of illness or mortality. The results are representative of six mice per condition. Survival percentage were statistically evaluated by the Log-rank Mantel-Cox test, ***($p<0.001$) and **($p<0.01$) (n=6). (**J**) Histopathological analysis of kidney sections using GMS staining was done for kidneys isolated from mice administrated with normal distilled water or NAC-containing distilled water and injected with *ΔΔpfk1*. Fungal cells appear black in colour against the colored background of the kidney tissue. Bright field imaging was done using a Zeiss Axioplan 2 microscope at 40X magnification. Scale bar represents 50 µm. This figure was created using Biorender.com.

(*Figure 5C*). Overall, this data indicates that the *ΔΔpfk1* strain exhibits significantly reduced survival compared to the wild-type, in the presence of macrophages.

Given that the *ΔΔpfk1* strain was significantly attenuated in the microbial survival assay, we next wanted to test the ability of this deletion strain to establish systemic infection in a host. In order to do this, we challenged healthy wild-type C57BL/6 mice intravenously with a dose of $1\times10^7$ colony-forming units (CFU) (lethal dose *Hirayama et al., 2020*) of *C. albicans* wild-type SC5314 or *ΔΔpfk1* strain via lateral tail vein injection. This is an established murine model of systemic candidiasis in the field and faithfully mimics the infection that occurs in humans (*Segal and Frenkel, 2018*). We then monitored the survival of the infected animals for 21 days. Mice challenged with *ΔΔpfk1* strain had significantly increased survival compared to mice challenged with *C. albicans* wild-type SC5314 (*Figure 5D*). *C. albicans* exhibits strong tropism towards kidneys during murine infections (*Lionakis et al., 2011*; *Pappas et al., 2018*) and in order to measure the fungal burden within the host, kidneys from C57BL/6 mice infected with both *C. albicans* wild-type SC5314 and *ΔΔpfk1* were harvested ~2 days post infection (when the mice injected with wild-type *C. albicans* SC5314 become moribund), homogenized and plated to measure the fungal burden by enumerating the colony-forming units (CFU) (*Figure 5E*). Our data indicates that kidneys isolated from mice challenged with the *ΔΔpfk1* strain had significantly lesser CFU compared to kidneys isolated from mice challenged with the *C. albicans* wild-type SC5314 (*Figure 5F*). We also performed histological analysis on the infected kidneys using GMS staining (Grocott Methenamine Silver staining) to visualize the fungi (stained black) in the tissue. Kidneys isolated from mice challenged with the *C. albicans* wild-type SC5314 had higher amounts of fungi compared to kidneys isolated from mice challenged with the *ΔΔpfk1* strain (*Figure 5G*). Overall, our data indicates that the perturbation of glycolysis significantly attenuates *C. albicans* virulence.

Our previous in vitro results clearly demonstrated that exogenous supplementation of cysteine and methionine significantly rescued the hyphal differentiation defect exhibited by the *ΔΔpfk1* strain. Based on this, we wanted to check whether sulfur supplementation could also rescue the virulence defects exhibited by *ΔΔpfk1*, in a murine model of systemic candidiasis. Given that N-acetyl cysteine (NAC) is a commonly used exogenous sulfur source for in vivo studies (*Shee et al., 2022*), we supplemented distilled water with NAC (6 mg/ml concentration) and this was administered to mice orally 72 hr prior to infection. After 72 hr of pre-treatment with NAC, we challenged healthy wild-type C57BL/6 mice intravenously with a dose of $1\times10^7$ CFU of wild-type SC5314 or *ΔΔpfk1* strain via lateral tail vein injection (*Figure 5H*). We then monitored the survival of the infected animals for 21 days. Remarkably, mice administered with NAC containing distilled water had significantly reduced survival compared to mice administered with normal distilled water (*Figure 5I*). We also performed histological analysis on

the infected kidneys using GMS staining to visualize the fungi in the tissue. Kidneys isolated from mice administered with NAC containing distilled water and infected with ΔΔpfk1 had significantly higher amounts of fungi compared to mice administered with normal distilled water (*Figure 5J*). This clearly demonstrates that exogenous sulfur supplementation to the host via NAC administration reverses the attenuated virulence exhibited by the ΔΔpfk1 strain.

## Discussion

Our findings provide compelling evidence for a conserved metabolic network that intricately links central carbon metabolism (particularly glycolysis), sulfur amino acid biosynthesis, and fungal morphogenesis in both *Saccharomyces cerevisiae* and *Candida albicans*. While nitrogen limitation is a well-established and essential trigger for the yeast to pseudohyphae transition in *S. cerevisiae*, the specific influence of carbon sources on this process has not been completely understood. Studies have shown that fermentable carbon sources are critical for inducing pseudohyphal differentiation under nitrogen-limiting conditions (*Van de Velde and Thevelein, 2008*). Conversely, when these fermentable sugars are replaced with non-fermentable carbon sources, a significant reduction in pseudohyphal differentiation is observed (*Strudwick et al., 2010*). These findings collectively suggest that the presence of fermentable carbon sources, and by extension their metabolism, is a critical requirement for inducing fungal morphogenesis in *S. cerevisiae*. Glucose in the extracellular environment can be sensed by *S. cerevisiae* using multiple proteins. One such protein is the well-characterized G-protein-coupled receptor (GPCR), Gpr1. Binding of glucose to Gpr1 activates a downstream signalling cascade, resulting in the production of cAMP which activates Protein Kinase A (PKA) (*Kraakman et al., 1999*; *Yun et al., 1997*). This glucose-dependent cAMP-PKA pathway has been shown to be essential for pseudohyphal differentiation under nitrogen-limiting conditions (*Lorenz et al., 2000*). Additionally, a key glycolytic intermediate, fructose-1,6-bisphosphate (FBP), is also known to activate the cAMP-PKA pathway via activation of Ras proteins, further linking glucose metabolism to this crucial signalling cascade (*Peeters et al., 2017*). However, the role of glucose as a metabolite in the context of pseudohyphal differentiation is not known. Building upon previous data that fermentable carbon sources that are metabolized via glycolysis are able to induce fungal morphogenesis, we asked whether the ability of these cells to undergo active glycolysis is critical for fungal morphogenesis. We took a two-pronged approach which involved pharmacological or genetic perturbation of glycolysis to address this question. Our findings clearly demonstrated that active glycolysis is essential for fungal morphogenesis under nitrogen-limiting conditions. To understand whether the pseudohyphal differentiation defects we observed due to glycolysis perturbation were solely due to the inability of glucose to activate glucose-dependent cAMP-PKA pathway, we performed cAMP add-back assays and interestingly, exogenous supplementation of cAMP failed to rescue pseudohyphal differentiation defects caused by the perturbation of glycolysis (through inhibitors (2DG and NaCi) or through genetic knockout strains (ΔΔpfk1 and ΔΔadh1)). Our results clearly demonstrate that the ability of *S. cerevisiae* to efficiently metabolize glucose via glycolysis is critical for pseudohyphal differentiation under nitrogen-limiting conditions, in a cAMP-PKA-independent manner. This implies that glycolysis may be parallelly regulating other cellular mechanisms essential for this morphological transition. It is plausible that the glycolytic flux, or a yet-to-be-identified downstream intermediate, acts as a distinct metabolic signal that is parallelly important for pseudohyphal differentiation along with the cAMP-PKA pathway. This uncovers a previously unrecognized layer of regulatory complexity in the interplay between central carbon metabolism and fungal morphogenesis.

To gain deeper insights underlying the glycolysis-mediated regulation of fungal morphogenesis in a cAMP-PKA-independent manner, we performed comparative transcriptomics on *S. cerevisiae* wild-type cells grown under nitrogen-limiting conditions with and without the glycolysis inhibitor, 2DG (which had the strongest inhibitory effect on pseudohyphal differentiation). As anticipated, genes involved in amino acid biosynthesis and transport were upregulated in the 2DG-treated cells, consistent with a feedback response that cells exhibit when glycolysis is perturbed under nitrogen-limiting conditions, as various intermediates of glycolysis are known to serve as precursors for the biosynthesis of several amino acids under these conditions (*Dikicioglu et al., 2011*; *Martíez-Force and Benítez, 1992*). Remarkably, we observed a striking and unexpected downregulation of multiple genes specifically involved in the biosynthesis and transport of sulfur-containing amino acids, cysteine and methionine, in the 2DG-treated cells at both early and late time points of pseudohyphal development. This

transcriptional downregulation was further validated at the protein level for key proteins involved in the sulfur assimilation pathway (Met4, Met32, Met16, Met10, Cys4, and Cys3), clearly demonstrating the dependence of sulfur metabolism on active glycolysis during fungal morphogenesis. This unexpected link between central carbon metabolism and sulfur metabolism is not restricted to conditions that induce fungal morphogenesis, i.e., in surface-attached colonies grown in nitrogen-limiting conditions, since even in liquid cultures that have limiting levels of nitrogen, wherein *S. cerevisiae* cells do not undergo pseudohyphal differentiation (*Gancedo, 2001*; *Gimeno et al., 1992*), perturbation of glycolysis with 2DG results in significantly reduced expression of proteins involved in the sulfur assimilation pathway (Met32, Met16, Met10, and Cys3).

Given that multiple genes that encode for proteins involved in de novo biosynthesis of sulfur-containing amino acids were downregulated in the presence of 2DG, which strongly inhibits fungal morphogenesis, we wanted to explore the possibility that perturbation of sulfur metabolism is causal for the attenuation of fungal morphogenesis in the presence of 2DG. We tested the functional relevance of this glycolysis-dependent regulation of sulfur metabolism in fungal morphogenesis directly through exogenous supplementation of sulfur sources. Interestingly, cysteine specifically rescued the pseudohyphal differentiation defect exhibited by *S. cerevisiae* in the presence of 2DG. However, methionine was unable to rescue the pseudohyphal differentiation defect in the presence of 2DG even at various different concentrations (50 μM, 100 μM, 200 μM, and 500 μM). The observed differential rescue effects of cysteine and methionine could be due to the distinct amino acid transport systems used by *S. cerevisiae* to transport these amino acids. *S. cerevisiae* primarily uses multiple, low-affinity permeases (Gap1, Bap2, Bap3, Tat1, Tat2, Agp1, Gnp1, and Yct1) for cysteine transport, while relying on a limited set of high-affinity transporters (like Mup1) for methionine transport, with the added complexity that its methionine transporters can also transport cysteine (*Düring-Olsen et al., 1999*; *Huang et al., 2017*; *Kosugi et al., 2001*; *Menant et al., 2006*). Hence, it is likely that cysteine uptake could be happening at a higher efficiency in *S. cerevisiae* compared to methionine uptake. Therefore, to achieve a comparable functional rescue by exogenous supplementation of methionine, it is necessary to use a higher concentration of methionine. When we performed our rescue experiments using higher concentrations of methionine, we did not see any rescue of pseudohyphal differentiation in the presence of 2DG, and in fact, we noticed that, at higher concentrations of methionine, the wild-type strain failed to undergo pseudohyphal differentiation even in the absence of 2DG. This is likely due to the fact that increasing the methionine concentration raises the overall nitrogen content of the medium, thereby making the medium less nitrogen-starved. This presents a major experimental constraint, as pseudohyphal differentiation is strictly dependent on nitrogen limitation, and the elevated nitrogen resulting from the higher methionine concentration can inhibit pseudohyphal differentiation. Similarly, gene expression analysis using RT-qPCR revealed a significant downregulation of multiple genes involved in the biosynthesis of sulfur-containing amino acids (*met32*, *met3, met5,* and *met17)* in ΔΔ*pfk1* and ΔΔ*adh1* strains and the pseudohyphal differentiation defects observed in these strains were completely rescued by the addition of cysteine or methionine. These findings strongly implicate that de novo biosynthesis of sulfur-containing amino acids is a critical downstream effector of glycolytic activity in promoting pseudohyphal differentiation under nitrogen limitation. To directly assess the role of sulfur metabolism in fungal morphogenesis, we generated and characterized the *met32* deletion strain. This particular strain was critical for our investigation because, unlike other *MET* gene deletions, *met32* deletion does not result in auxotrophy for methionine or cysteine. This unique characteristic is attributed to the functional redundancy provided by its paralog, *met31* (*Blaiseau et al., 1997*). The deletion of *met32* resulted in substantial defects in pseudohyphal differentiation under nitrogen-limiting conditions, directly implicating the role of sulfur metabolism in regulating fungal morphogenesis. Our sulfur add-back experiments demonstrated that cysteine or methionine rescued the morphogenetic defects exhibited by the *met32* knockout strain, further solidifying the direct involvement of sulfur metabolism in this morphogenetic switching process. How active sulfur metabolism contributes towards fungal morphogenesis is not yet understood, but given its critical requirement during this phenomenon, it would be an important question to explore and will further our overall understanding of this important biological phenomenon.

It is interesting to note that the expression of Met4, the principal transcriptional regulator of the de novo biosynthesis of sulfur-containing amino acids in *S. cerevisiae,* along with Met32, one of its cognate DNA-binding cofactors (*Blaiseau et al., 1997*), are significantly downregulated when

glycolysis is perturbed. Given the obligate requirement of Met4 interaction with DNA-binding proteins, such as Met28, Met32, or Met31, to elicit transcriptional activation of the genes involved in the de novo biosynthesis of sulfur-containing amino acids (*Blaiseau and Thomas, 1998*), the observed downregulation in both Met4 and Met32 levels likely compromises the overall biosynthesis of sulfur-containing amino acids. This concurrent downregulation could explain the attenuated expression of downstream genes involved in the synthesis of methionine and cysteine. This suggests a novel regulatory mechanism wherein the expression of Met4/Met32 is coupled to glycolytic activity under nitrogen-limiting conditions. Given that sulfur metabolism is coupled to glycolysis in *C. albicans* as well and contributes towards fungal morphogenesis, it is likely that this conserved regulatory axis might be critical for fungal morphogenesis in a broad range of fungal species. Interestingly, Met4 regulation is directly dependent on Met30 (*Rouillon et al., 2000*). Met30 functions as a molecular switch for Met4, the master regulator of the *MET* gene network. Under sulfur-replete conditions, Met30 ensures that Met4 remains inactive through ubiquitination and proteasomal degradation. Conversely, a reduction in sulfur availability diminishes Met30 activity, thereby de-repressing Met4 and allowing it to induce the expression of genes necessary for sulfur amino acid biosynthesis (*Rouillon et al., 2000*; *Smothers et al., 2000*; *Thomas and Surdin-Kerjan, 1997*). Given this regulatory mechanism, we investigated the expression of proteins involved in the sulfur assimilation pathway (Met4, Met32, Met16, and Cys3), in a viable Δmet30 deletion strain, under conditions where glycolysis was perturbed using 2DG. We observed no difference in protein levels of Met4, Met32, Met16, and Cys3, between 2DG-treated and untreated cells. This indicates that, in the heterozygous deletion strain of *met30* (*Δmet30*), Met4 is not degraded even when glycolysis is perturbed, supporting the hypothesis that perturbing active glycolysis with inhibitors like 2DG increases Met30 activity, which in turn leads to the increased degradation of Met4, which would then negatively affect *MET* gene transcription, subsequently attenuating pseudohyphal differentiation. However, a surprising finding emerged when we examined Met30 levels in the presence of 2DG. There was no increase in the levels of Met30 in 2DG-treated cells compared to untreated cells. This suggests that glycolysis influences Met30 activity through a post-translational mechanism, rather than by altering its protein abundance. Studies have shown that under cadmium stress, Met4 activity is upregulated by reducing the activity of Met30 through post-translational events, specifically by inducing the dissociation of Met30 from the core SCF (Skp1/Cullin1/F-box) complex without affecting Met30 protein levels (*Lauinger et al., 2024*; *Yen et al., 2005*). This well-established paradigm provides a compelling framework for our observation, suggesting that active glycolytic flux, similar to cadmium stress, might rapidly modulate the activity of Met30 by decreasing its association with the SCF complex to control Met4 activity, rather than through changes in protein synthesis or degradation. Future studies will be needed to delineate the precise post-translational modifications or regulatory interactions that modulate Met30 activity in response to glycolytic flux.

Intriguingly, our work extended these findings to the pathogenic fungus *C. albicans*, revealing a conserved role for glycolysis-mediated sulfur metabolism in hyphal differentiation, a key virulence attribute in this species (*Gow et al., 2011*; *Moyes et al., 2016*; *Wilson et al., 2016*). Similar to our observations in *S. cerevisiae*, *C. albicans* exhibited glycolysis-dependent hyphal formation under nitrogen limitation in a cAMP-PKA-independent manner. Perturbation of glycolysis (either by 2DG addition or the deletion of *pfk1*) resulted in the significant downregulation of genes involved in de novo biosynthesis of sulfur-containing amino acids (*met32, met3, met5* (*ecm17*), *met10* and *met17* (*met15*)) and the hyphal differentiation defects that occur in response to glycolysis perturbation could be rescued by the exogenous supplementation of cysteine or methionine, whereas the exogenous addition of cAMP failed to rescue the hyphal defects caused by the addition of 2DG or the deletion of *pfk1*. These results clearly demonstrate that de novo biosynthesis of sulfur-containing amino acids is a critical downstream effector of glycolytic activity in promoting hyphal differentiation under nitrogen-limiting conditions in *C. albicans* as well. Given that central carbon metabolic pathways like glycolysis and sulfur assimilation pathway are broadly conserved metabolic processes across a plethora of fungal species, it is possible that this novel regulatory axis might play a crucial role in modulating fungal morphogenesis in multiple fungal species, and further studies are warranted to explore this. Finally, given the established link between hyphal differentiation and *C. albicans* virulence, we explored the

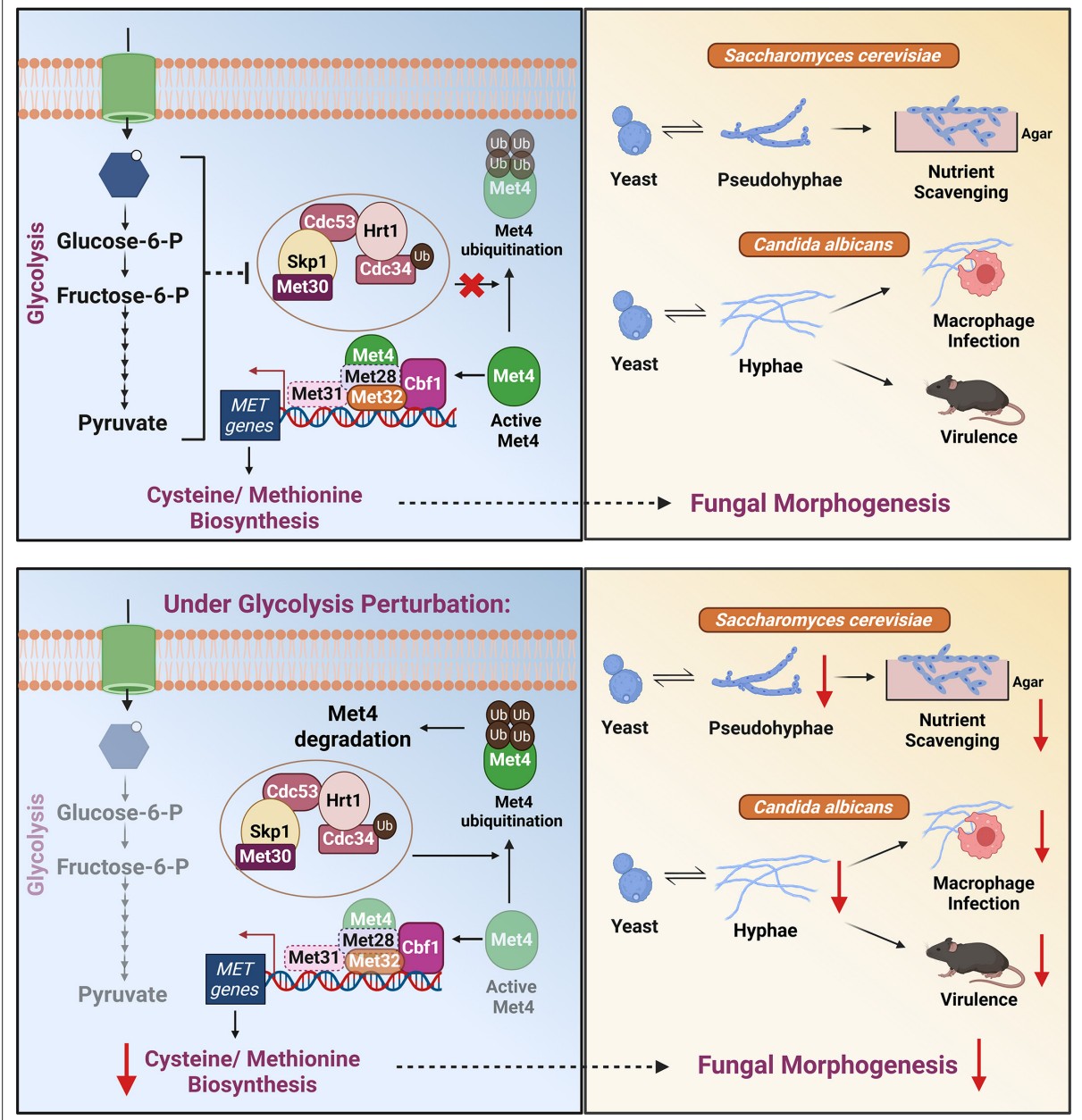

**Figure 6.** Glycolysis-dependent sulfur metabolism orchestrates morphological transitions and virulence in fungi. Active glycolysis plays a key regulatory role in the de novo biosynthesis of sulfur-containing amino acids by modulating the activity of SCF$^{Met30}$ complex (SCF$^{Met30}$ complex includes Met30, Skp1, Cdc53, Hrt1, and Cdc34) which in turn affects the expression of genes involved in this process (In *S. cerevisiae*, *MET* transcription complex includes Met4, Met28, Cbf1, Met31, and Met32, whereas in *C. albicans*, the characterized components of this complex include Met4, Cbf1, and Met32. Hence, Met28 and Met31, unique to the *MET* transcription complex of *S. cerevisiae* are represented with dotted borders). Glycolysis-dependent sulfur metabolism, in turn, is critical for fungal morphogenesis in both species and the ability of *C. albicans* to survive within macrophages and cause systemic infection in a murine model of candidiasis. Perturbation of active glycolysis increases Met30 activity, which leads to the increased degradation of Met4, resulting in the reduced expression of genes involved in the de novo biosynthesis of sulfur-containing amino acids. This, in turn, attenuates fungal morphogenesis in both species and the ability of *C. albicans* to survive within macrophages and cause systemic infection in a murine model of candidiasis. This figure was created using Biorender.com.

impact of glycolytic perturbation on the pathogenicity of this fungus. Interestingly, RT-qPCR analysis revealed a significant downregulation of genes involved in hyphal differentiation and virulence (*als3*, *ece1*, *hwp1*, *hyr1*, *ihd1*, *rbt1*, and *sap6*) upon 2DG treatment and in the ΔΔ*pfk1* strain. Interestingly, deletions of sugar kinases or transcription factors regulating the expression of glycolysis genes or *adh1*

in *C. albicans* leads to filamentation defects and downregulation of filamentation/virulence-specific genes (*Askew et al., 2009*; *Laurian et al., 2019*; *Song et al., 2019*). Our data, showing transcriptional downregulation of hyphal differentiation and virulence genes following glycolytic perturbation via the deletion of *pfk1*, strongly corroborate these previous findings. This collectively positions glycolysis as a central metabolic nexus, rather than merely a catabolic process, profoundly influencing the pathogenic ability of *C. albicans* by intricately linking cellular metabolism, morphogenetic transitions, and the expression of its virulence factors. Furthermore, the *ΔΔpfk1* strain displayed significantly reduced survival within murine macrophages and attenuated virulence in a murine model of systemic candidiasis, as evidenced by increased host survival, lower kidney fungal burden, and reduced tissue colonization. Remarkably, exogenous sulfur supplementation (N-acetyl cysteine (NAC)) to mice rescued the attenuated virulence exhibited by *ΔΔpfk1*. It is important to note that, in the context of certain bacterial pathogens, NAC has been reported to augment cellular respiration, subsequently increasing reactive oxygen species (ROS) generation, which contributes to pathogen clearance (*Shee et al., 2022*). Interestingly, in our study, NAC supplementation to the mice was given prior to the infection and maintained continuously throughout the duration of the experiment. This continuous supply of NAC likely contributes to the rescue of virulence defects exhibited by the *ΔΔpfk1* strain. Essentially, NAC likely allows the mutant to fully activate its essential virulence strategies (including morphological switching) to cause a successful infection in the host. These findings underscore the critical role of glycolysis, and by extension, glycolysis-dependent sulfur metabolism, in the virulence of *C. albicans*.

In conclusion, our study unveils a conserved metabolic network that orchestrates fungal morphogenesis in response to nutrient availability. We demonstrate that active glycolysis, fueled by fermentable carbon sources in a cAMP-PKA-independent manner, plays a crucial role in regulating the de novo biosynthesis of sulfur-containing amino acids in Met30-dependent manner, which in turn are essential for the morphological transitions in both *S. cerevisiae* and *C. albicans* (*Figure 6*). The attenuation of *C. albicans* virulence upon disruption of this pathway highlights the potential of targeting specific fungal metabolic networks as a novel antifungal strategy. Future research in our laboratory will focus on elucidating the precise molecular mechanisms by which glycolytic intermediates or downstream signals regulate the expression of sulfur metabolism pathway and how sulfur-containing amino acids contribute to the cellular processes underlying fungal morphogenesis.

# Materials and methods

## Key resources table

| Reagent type (species) or resource | Designation | Source or reference | Identifiers | Additional information |
|---|---|---|---|---|
| Recombinant DNA reagent | pFA6a-kanMX | Gift from Dr. Sunil Laxman | | |
| Recombinant DNA reagent | pFA6a-Hyg | Gift from Dr. Sunil Laxman | | |
| Recombinant DNA reagent | pFA6a-Nat | Gift from Dr. Sunil Laxman | | |
| Recombinant DNA reagent | pFA6a-Ctag-HA-G418 | Gift from Dr. Sunil Laxman | | |
| Recombinant DNA reagent | pFA6a-Ctag-HA-NAT | Gift from Dr. Sunil Laxman | | |
| Recombinant DNA reagent | pSFS2a-NAT | Gift from Dr. Kaustuv Sanyal | | |
| Antibody | Rabbit anti-HA | Cell Signaling Technology | Cat. #3724S | (1:2000 dilution) |
| Antibody | Mouse anti-β-actin | Cell Signaling Technology | Cat. #3700S | (1:2000 dilution) |
| Antibody | Mouse anti-PGK1 | Santa Cruz Biotechnology | Cat. #sc-130335 | (1:2000 dilution) |
| Antibody | Goat anti-Rabbit IgG-HRP conjugate | Cell Signaling Technology | Cat. #7074S | (1:4000 dilution) |
| Antibody | Horse anti-Mouse IgG-HRP conjugate | Cell Signaling Technology | Cat. #7076S | (1:4000 dilution) |
| Chemical compound, drug | BD Bacto agar | Becton Dickinson | Cat. #214010 | |
| Chemical compound, drug | Agarose | MP Biomedicals | Cat. #100267 | |
| Chemical compound, drug | Ammonium sulfate | Sisco Research Laboratories | Cat. #0149175 | |

*Continued on next page*

*Continued*

| Reagent type (species) or resource | Designation | Source or reference | Identifiers | Additional information |
|---|---|---|---|---|
| Chemical compound, drug | Blotto, non-fat dry milk | Santa Cruz Biotechnology | Cat. #sc-2324 | |
| Chemical compound, drug | β-mercaptoethanol | Sigma-Aldrich | Cat. #444203 | |
| Chemical compound, drug | Ethanol | HiMedia | Cat. #MB228 | |
| Chemical compound, drug | D-(+)-Glucose anhydrous | HiMedia | Cat. #GRM016 | |
| Chemical compound, drug | Lithium acetate | Sigma-Aldrich | Cat. #517992 | |
| Chemical compound, drug | Peptone | Thermo Fisher Scientific | Cat. #211677 | |
| Chemical compound, drug | RNaseZAP | Sigma-Aldrich | Cat. #R2020 | |
| Chemical compound, drug | SDS | G Biosciences | Cat. #RC1184 | |
| Chemical compound, drug | Trichloroacetic acid | Sigma-Aldrich | Cat. #100807 | |
| Chemical compound, drug | Tris, free base | HiMedia | Cat. #MB029 | |
| Chemical compound, drug | Tween 20 | Sigma-Aldrich | Cat. #P7949 | |
| Chemical compound, drug | Yeast extract | Thermo Fisher Scientific | Cat. #212750 | |
| Chemical compound, drug | Yeast nitrogen base w/o amino acids and ammonium sulfate | Becton Dickinson | Cat. #233520 | |
| Chemical compound, drug | L-Cysteine | Sigma-Aldrich | Cat. #30089 | |
| Chemical compound, drug | L-Methionine | Sigma-Aldrich | Cat. #64319 | |
| Chemical compound, drug | Sodium citrate | Sigma-Aldrich | Cat. #C8532 | |
| Chemical compound, drug | Glycerol | HiMedia | Cat. #MB060 | |
| Chemical compound, drug | Potassium acetate | MP Biomedicals | Cat. #191425 | |
| Chemical compound, drug | 2-Deoxy-D-Glucose | Sigma-Aldrich | Cat. #D8375 | |
| Chemical compound, drug | Nourseothricin | Jena Bioscience | Cat. #AB-102XL | |
| Chemical compound, drug | G418 | MP Biomedicals | Cat. #158782 | |
| Chemical compound, drug | Ethidium bromide | HiMedia | Cat. #MB074 | |
| Chemical compound, drug | Pico-ECL substrate | Thermo Fisher Scientific | Cat. #34579 | |
| Chemical compound, drug | Femto-ECL substrate | Thermo Fisher Scientific | Cat. #34094 | |
| Chemical compound, drug | TEMED | Sisco Research Laboratories | Cat. #52145 | |
| Chemical compound, drug | Hygromycin | Sigma-Aldrich | Cat. #H0654 | |
| Commercial assay or kit | Ribopure RNA purification Kit-Yeast | Thermo Fisher Scientific | Cat. #AM1926 | |
| Commercial assay or kit | BCA Protein Assay Kit | Thermo Fisher Scientific | Cat. #23227 | |
| Commercial assay or kit | PrimeScript cDNA Synthesis Kit | Takara | Cat. #6110A | |
| Commercial assay or kit | GMS Staining Kit | ABcam | Cat. #AB287884 | |
| Software, algorithm | RStudio v4.3.2 | Posit PBC | | |
| Software, algorithm | GraphPad Prism v9 | GraphPad Software | | |
| Software, algorithm | ImageJ | Open-source Software | | |
| Software, algorithm | IGV v2.16.2 | Open-source Software | | |

## Yeast strains

The prototrophic diploid strains, ∑1278b or CEN.PK were used in all the experiments involving *S. cerevisiae*. The prototrophic diploid strain, SC5314, was exclusively used in all the experiments involving *C. albicans*. The yeast strains used in this study are listed in *Supplementary file 1*.

## Generation of *S. cerevisiae* gene knockout and epitope-tagged strains

A standard PCR-based technique was used to amplify resistance cassettes (G418, HygB, or NAT) with flanking sequences in order to carry out gene deletions. The target gene was then replaced by homologous recombination using the lithium acetate-based transformation (*Schiestl and Gietz, 1989*). Similarly, C-terminal epitope-tagged yeast strains were generated using a PCR-based technique. This involved amplifying resistance cassettes, flanked by sequences homologous to the C-terminal region of the target gene, along with the desired epitope tags. The targeted gene was then tagged at its C-terminus through homologous recombination, using the lithium acetate-based transformation method. Primers used for these experiments are listed in *Supplementary file 2*.

## Generation of *C. albicans* gene knockout strains

The *C. albicans* strains used in this study are listed in *Supplementary file 1*. The previously established SAT1-flipper method was used with some modifications for the deletion of the desired gene (*Reuss et al., 2004*). To generate the *pfk1* deletion (*ΔΔpfk1*) strain, a DNA fragment carrying the flanking regions of *pfk1* (100 bps homology) and the *SAT1*-flipper cassette was transformed into *C. albicans* wild-type SC5314, using electroporation. Transformants were selected based on their nourseothricin (NAT) resistance. PCR confirmation was used to validate the transformants. The *SAT1*-flipper cassette was then excised from the *pfk1* locus by growing the cells in YPM medium (10 g/l yeast extract, 20 g/l peptone, and 2% (w/v) maltose) to induce expression of the *MAL2* promoter-regulated recombinase. To knock out the second allele of *pfk1*, the heterozygous *pfk1* deletion mutants (*Δpfk1/PFK1)* were used for the transformation wherein the DNA fragment carrying the flanking regions of *pfk1* (100 bps homology) and the *SAT1*-flipper cassette was transformed into *C. albicans Δpfk1/PFK1* SC5314. Transformants were selected based on their NAT resistance. PCR confirmation was used to validate the transformants. Primers used to generate or confirm the deletion strains are listed in *Supplementary file 2*.

## Media and growth conditions

YPD broth containing 10 g/l yeast extract, 20 g/l peptone, and 2% (w/v) glucose was used for the overnight growth of the cultures. Nitrogen-limiting medium containing 1.7 g/l yeast nitrogen base without amino acids or ammonium sulfate, 2% (w/v) glucose (SLAD), 50 µM ammonium sulfate and 2% (w/v) agar which was extensively washed, to remove excess nitrogen (*Gimeno et al., 1992*) was used for all experiments involving pseudohyphal or hyphal differentiation. Overnight cultures of various strains were grown in YPD broth and incubated at 30 °C with continuous shaking at 200 rotations per minute (RPM). Reagents and software used in this study are mentioned in the Key resources table.

## Pseudohyphal differentiation assay

To induce pseudohyphal differentiation in *S. cerevisiae,* overnight cultures of various strains (listed in *Supplementary file 1*) were spotted on nitrogen-limiting medium (SLA) containing 2% (w/v) glucose (SLAD). In order to check the effect of glycolysis inhibitors, 0.05% (w/v) concentration of 2-Deoxy-D-Glucose (2DG), and 0.5% (w/v) concentration of sodium citrate (NaCi), on pseudohyphal differentiation, wild-type was spotted on SLAD containing sub-inhibitory concentrations of 2DG (0.05% w/v) or NaCi (0.5% w/v). Plates were incubated for 10 days at 30 °C (*Chandarlapaty and Errede, 1998*; *González et al., 2017*). After 10 days, imaging was done using an Olympus MVX10 stereo microscope. Cells from colonies were isolated and single cells were imaged using a Zeiss Apotome microscope. The length/width ratio of individual cells was measured using ImageJ, and the percentage of pseudohyphal cells from the total population was determined. Cells with a length/width ratio of 2 or more were considered as pseudohyphal cells as described previously (*Schröder et al., 2000*). Statistical analysis was done using unpaired t-test or one-way ANOVA test. Error bars represent SEM. All the experiments were performed independently, thrice.

## Hyphal differentiation assay

To induce hyphal differentiation in *C. albicans*, overnight cultures of each strain (listed in *Supplementary file 1*) were spotted on nitrogen-limiting medium (SLAD). To check the effect of the glycolysis inhibitor, 2DG, on hyphal differentiation, wild-type was spotted on SLAD containing sub-inhibitory concentration of 2DG (0.2% w/v). All the plates were incubated for 7 days at 37 °C (*Sánchez-Martínez*

*and Pérez-Martín, 2002*; *Song et al., 2019*). After 7 days, cells from colonies were isolated and single cells were imaged using a Zeiss Apotome microscope. The length/width ratio of individual cells was measured using ImageJ, and the percentage of hyphal cells from the total population was determined. Cells with a length/width ratio of 4.5 or more were considered as hyphal cells as described previously (*Su et al., 2018*). Statistical analysis was done using unpaired t-test or one-way ANOVA test. Error bars represent SEM. All the experiments were performed independently, thrice.

### Growth curves

Growth curve analysis in the presence of glycolysis inhibitors (2DG or NaCi) was performed as follows. Overnight grown cultures of wild-type strains (∑1278b or CEN.PK or SC5314) were diluted to $OD_{600}$=0.01 in fresh SLAD medium with and without 2DG or NaCi and allowed to grow at 30 °C for 24 hr. $OD_{600}$ was recorded at 3 hr intervals. Growth curve analysis using various deletion strains were performed as follows. Overnight grown cultures of various deletion strains (listed in *Supplementary file 1*) along with wild-type strains (∑1278b or CEN.PK or SC5314) were diluted to $OD_{600}$=0.01 in fresh SLAD and allowed to grow at 30 °C for 24 hr. $OD_{600}$ was recorded at 3 hr intervals. Graphs were prepared using GraphPad Prism. Error bars represent SEM. All the experiments were performed independently, thrice.

### cAMP add-back assays

In order to perform cAMP add-back assays, overnight cultures of wild-type ∑1278b and various deletion strains of *S. cerevisiae* (listed in *Supplementary file 1*) were spotted on SLAD or SLAD containing sub-inhibitory concentration of 2DG or NaCi, in the presence and absence of cAMP (1 mM) (*Lorenz and Heitman, 1997*). Plates were incubated for 10 days at 30 °C. After 10 days, imaging was done using an Olympus MVX10 stereo microscope. Cells from colonies were isolated and single cells were imaged using a Zeiss Apotome microscope. The length/width ratio of individual cells was measured using ImageJ and the percentage of pseudohyphal cells from the total population was determined. For hyphal differentiation assays in *C. albicans*, overnight cultures of wild-type SC5314 and *ΔΔpfk1* strains of *C. albicans* were spotted on SLAD or SLAD containing sub-inhibitory concentration of 2DG, in the presence and absence of cAMP (5 mM) (*Li et al., 2013*). Plates were incubated for 7 days at 37 °C. After 7 days, cells from colonies were isolated and single cells were imaged using a Zeiss Apotome microscope. The length/width ratio of individual cells was measured using ImageJ and the percentage of hyphal cells from the total population was determined. Statistical analysis was done using one-way ANOVA test. Error bars represent SEM. All the experiments were performed independently, thrice.

### Sulfur add-back assays

In order to perform sulfur add-back assays, overnight cultures of various strains of *S. cerevisiae* or *C. albicans* (listed in *Supplementary file 1*) were spotted on SLAD or SLAD containing sub-inhibitory concentration of 2DG, in the presence and absence of sulfur-containing compounds, including cysteine, methionine. Plates were incubated for 10 days at 30 °C to induce pseudohyphal differentiation in *S. cerevisiae*. After 10 days, imaging was done using an Olympus MVX10 stereo microscope. Cells from colonies were isolated and single cells were imaged using a Zeiss Apotome microscope. The length/width ratio of individual cells was measured using ImageJ and the percentage of pseudohyphal cells from the total population was determined. For hyphal differentiation assays in *C. albicans*, plates were incubated for 7 days at 37 °C. After 7 days, cells from colonies were isolated and single cells were imaged using a Zeiss Apotome microscope. The length/width ratio of individual cells was measured using ImageJ and the percentage of hyphal cells from the total population was determined. Statistical analysis was done using unpaired t-test or one-way ANOVA test. Error bars represent SEM. All the experiments were performed independently, thrice.

### Whole genome sequencing
#### Sample collection

Strains of *S. cerevisiae* (*ΔΔpfk1* or *ΔΔadh1* along with wild-type strain ∑1278b) or *C. albicans* (*ΔΔpfk1* along with wild-type strain SC5314) were grown in YPD medium at 30 °C for overnight.

## Genomic DNA extraction

Genomic DNA from the samples was extracted using the phenol-chloroform method. Following RNAse treatment to eliminate any contaminating RNA from the sample, the extracted DNA was submitted to the CCMB next-generation sequencing facility for whole genome sequencing.

## Whole genome sequencing analysis

Raw sequencing reads were obtained in FASTQ format. The reference genome and annotation files for the *S. cerevisiae* strain ∑1278b were downloaded from the *Saccharomyces* Genome Database (SGD) and reference genome and annotation files for the *C. albicans* strain SC5314 were downloaded from the NCBI (National Center for Biotechnology Information). Quality assessment of the raw reads was performed using FastQC (version 0.12.1). Adapter sequences were trimmed using Cutadapt (version 4.6). The reference genome was indexed with Hisat2 (version 2.2.1), and Samtools (version 1.20) was used to filter out multi-mapped reads and convert SAM files to BAM format. Visualization of reads was done using IGV software (version 2.16.2).

## RNA sequencing

### Sample collection

Overnight culture of *S. cerevisiae* wild-type ∑1278b was spotted on nitrogen-limiting media containing 2% (w/v) glucose (SLAD) with and without sub-inhibitory concentration of 2DG. The plates were incubated at 30 °C and colonies were isolated on day 5 (D-5) and day 10 (D-10).

### RNA-extraction

Total RNA from the sample was extracted using the protocol mentioned in the Ribopure RNA purification kit (yeast) (Cat. #AM1926). Following a DNAse treatment to eliminate any genomic DNA from the sample, the extracted RNA was submitted to the CCMB next-generation sequencing facility for RNA-Seqencing. The NovaSeq 6000 equipment was used to sequence transcriptomes.

### RNA-sequencing analysis

Raw sequencing reads were obtained in FASTQ format. The reference genome and annotation files for the *S. cerevisiae* strain ∑1278b were downloaded from the *Saccharomyces* Genome Database (SGD). Quality assessment of the raw reads was performed using FastQC (version 0.12.1). Adapter sequences were trimmed using Cutadapt (version 4.6). The reference genome was indexed with Hisat2 (version 2.2.1), and Samtools (version 1.20) was used to filter out multi-mapped reads and convert SAM files to BAM format. Read counting for uniquely mapped reads was done with FeatureCounts, using gene annotation data from the reference *S. cerevisiae* ∑1278b genome. Normalization and differential gene expression analysis were performed using R (version 4.3.2). Genes with a $\log_2$ fold change of $\geq 1$ were considered to be upregulated, indicating at least a doubling in expression under the given condition, while those with a $\log_2$ fold change of $\leq -1$ were considered to be downregulated.

## Protein isolation

For Western blotting experiments, in order to isolate proteins from liquid cultures, overnight yeast cultures were back-diluted in 10 ml of liquid SLAD in the presence and absence of 2DG. The yeast cells were grown till an $OD_{600}$ of ~1 and cells were harvested by centrifugation at 3000 rpm for 10 min at room temperature (RT). To isolate proteins from colonies, overnight cultures of pertinent strains were spotted on SLAD with and without sub-inhibitory concentration of 2DG (0.05% (w/v)) and colonies were isolated and resuspended in 1 ml of 1X PBS (Phosphate Buffered Saline) after 5 days (D-5). Cells were harvested by centrifugation at 14,000 rpm for 10 min at room temperature. Pelleted cells (from liquid cultures and colonies) were resuspended in 300 µl of ice-cold 10% (w/v) TCA solution and disrupted using bead-beating by addition of 150 µl of glass beads at 4 °C. Supernatant was transferred to another tube. Next, proteins were pelleted by centrifugation at 14,000 rpm for 10 min at 4 °C and the supernatant was discarded. Each pellet was resuspended in 400 µl of SDS/glycerol buffer (7.3% (w/v) SDS, 29% (v/v) glycerol, and 83.3 mM tris-base) and boiled at 95 °C for 10 min with occasional vortexing followed by centrifugation at 14,000 rpm for 10 min at room temperature. Supernatant was transferred to a fresh tube and used for protein estimation using the BCA protein estimation

kit (Cat. #23227). Finally, all samples were diluted in SDS-PAGE loading buffer (100 mM Tris-Cl, 4% (w/v) SDS, 0.02% (w/v) bromophenol blue, 20% (w/v) glycerol, and 7.5% (v/v) β-mercaptoethanol) and heated for 5 min at 90 °C.

## Immunodetection
Protein samples were separated using SDS-PAGE on polyacrylamide gels and transferred to PVDF membranes with a Mini Trans-Blot Cell module (Bio-Rad). The membranes were blocked for 1 hr at room temperature with 5% (w/v) non-fat dry milk in 1X TBST buffer (20 mM Tris-Base, pH 7.6, 150 mM NaCl, 0.1% (v/v) Tween-20) with gentle orbital shaking. After blocking, the membranes were incubated overnight at 4 °C with primary antibodies (1:2000 dilution) in 1X TBST buffer containing 5% (w/v) non-fat dry milk. Following this, the membranes were washed three times for 10 min in 1X TBST and then incubated for 1 hr at room temperature with HRP-conjugated secondary antibodies (1:4000 dilution) in 1X TBST with 5% (w/v) non-fat dry milk. After three additional washes in 1X TBST, chemilumines-cent substrate was added, and bands were detected using the SuperSignal West chemiluminescent substrate (Cat. #34579, Cat. #34094) and visualized using the Bio-Rad ChemiDoc MP imaging system.

## Antibodies
HA-Tag (C29F4) Rabbit mAb (Cat. #3724S), β-actin (8H10D10) Mouse mAb (Cat. #3700S), Goat anti-Rabbit IgG-HRP conjugate (Cat. #7074S), and Horse anti-Mouse IgG-HRP conjugate (Cat. #7076S) were purchased from Cell Signaling Technology. Pgk1 Mouse mAb (Cat. #sc-130335) was purchased from Santa Cruz Biotechnology.

## Densitometric analysis
Raw images of Western blots were analyzed using ImageJ to normalize target protein expression with the expression of housekeeping proteins, including PGK1 or β-actin in order to make densitometric graphs. Statistical analysis was done using unpaired t-test, with error bars representing SEM. All the experiments were performed independently, thrice.

## Reverse transcription-quantitative polymerase chain reaction (RT-qPCR) assay
For *S. cerevisiae*, overnight cultures of ΔΔpfk1 and ΔΔadh1 along with wild-type strain Σ1278b were spotted on SLAD and plates were incubated at 30 °C for ~4 days. After 4 days, colonies were isolated and total RNA was extracted using the protocol mentioned in the Ribopure RNA purification kit, yeast (Cat. #AM1926). cDNA synthesis was carried out using PrimeScript cDNA synthesis kit (Cat. #6110A). Primers used for RT-qPCR were designed using the PrimeQuest tool of Integrated DNA Technologies (IDT). RT-qPCR was performed using iTaq Universal SYBR Green mastermix (Cat. #1725121). Actin was used as a housekeeping gene for normalization. The results were analyzed using the $2^{-\Delta\Delta CT}$ method (*Winer et al., 1999*) and statistical analysis was done using one-way ANOVA test, with error bars representing SEM. All the experiments were performed independently, thrice. Primers used in the RT-qPCR experiments are mentioned in *Supplementary file 2*.

For *C. albicans*, overnight cultures of *C. albicans* wild-type SC5314 was spotted on SLAD with and without sub-inhibitory concentration of 2DG (0.2% (w/v)) or *C. albicans* ΔΔpfk1 strain along with wild-type SC5314 was spotted on SLAD and plates were incubated at 37 °C for 4 days. After 4 days, colonies were isolated and total RNA was extracted using the protocol mentioned in the Ribopure RNA purification kit, yeast (Cat. #AM1926). cDNA synthesis was carried out using PrimeScript cDNA synthesis kit (Cat. #6110A). Primers used for RT-qPCR were designed using the PrimeQuest tool of Integrated DNA Technologies (IDT). RT-qPCR was performed using iTaq Universal SYBR Green mastermix (Cat. #1725121). Actin was used as a housekeeping gene for normalization. The results were analyzed using the $2^{-\Delta\Delta CT}$ method (*Winer et al., 1999*), and statistical analysis was done using one-way ANOVA test, with error bars representing SEM. All the experiments were performed independently, thrice. Primers used in the RT-qPCR experiments are mentioned in *Supplementary file 2*.

## In vitro microbial survival assay
In vitro microbial survival assay in macrophages was conducted as described in previous studies with some modifications (*Netea et al., 2002; Pradhan et al., 2016*). *RAW 264.7* macrophage cells used

in our study were obtained from the CSIR-CCMB Tissue Culture Facility after authenticating cell identity using STR profiling and confirming that mycoplasma contamination testing was negative. *RAW 264.7* macrophages ($1\times10^6$ cells) were cultured in six-well plates with DMEM supplemented with 10% FBS. After 16 hr of incubation, the cells were treated with *C. albicans* wild-type SC5314 or *ΔΔpfk1* at a multiplicity of infection (MOI) of 10, for 1 hr in antibiotic-free DMEM containing 10% FBS. The media was then removed, and the cells were washed six times with 1X PBS to eliminate any free, non-phagocytosed fungal cells. Next, the cells were lysed with 350 µl of lysis buffer containing 0.025% SDS in 1X PBS. The lysate volume was adjusted to 1 ml with 1X PBS, and a 100 µl undiluted aliquot was plated on YPD agar plate. Serial dilutions of the lysate were also plated on YPD agar plates. These plates were incubated overnight at 37 °C. The following day, CFU were counted and plotted. The percentage of survival was calculated as the number of CFU in the presence of macrophages divided by the number of CFU in the absence of macrophages. Statistical analysis was done using unpaired t-test, with error bars representing SEM. All the experiments were performed independently, thrice.

### In vivo murine model of systemic candidiasis

In vivo infection experiments were performed as previously described (*Majer et al., 2012*). Briefly, *C. albicans* wild-type SC5314 and *ΔΔpfk1* were grown overnight in YPD medium at 30 °C. The overnight cultures were back-diluted in 10 ml of YPD liquid medium and allowed to grow till OD of ~0.6, then washed and resuspended in 1X PBS. C57BL/6 wild-type mice, aged 6–8 weeks, were injected intravenously via lateral tail vein with *C. albicans* wild-type SC5314 ($1\times10^7$ CFU) or *ΔΔpfk1* ($1\times10^7$ CFU). After infection, mice were monitored for survival and daily for signs of morbidity and mortality were recorded over a period of 21 days. Survival data were analyzed using the Log-rank Mantel-Cox test. In order to perform in vivo sulfur supplementation, N-acetyl cysteine (NAC) was dissolved in distilled water at a concentration of 6 mg/ml and administered orally to mice before 72 hr of *C. albicans* infection. Normal distilled water administration to mice was used as a control. After 72 hr of NAC administration, C57BL/6 wild-type mice, aged 6–8 weeks, were infected with SC5314 ($1\times10^7$ CFU) or *ΔΔpfk1* strain ($1\times10^7$ CFU), intravenously via lateral tail vein injection. After infection, mice were monitored for survival and daily for signs of morbidity and mortality were recorded over a period of 21 days. Survival data were analyzed using the Log-rank Mantel-Cox test.

### Determination of fungal burden

To assess fungal burden, mice injected with *C. albicans* wild-type SC5314 or *ΔΔpfk1* were sacrificed at ~2 days and kidneys were harvested from infected mice. Kidneys were weighed and then homogenized using a sterilized and 1X PBS-washed Dounce homogenizer (Cat. #D8938). Kidney suspensions were then diluted in 1 ml of 1X PBS and serially diluted. Then 100 µl of dilutions was plated on YPD agar plates and allowed to grow at 37 °C. CFU were calculated by the following formula: (number of colonies × dilution factor × 10)/organ weight in grams (g). Statistical analysis was done using unpaired t-test. Error bars represent SEM. All the experiments were performed independently, thrice.

### Histopathology

For histopathological examination, kidneys were collected and fixed in 10% neutral buffered formalin for about 2 weeks. After fixation, the kidneys were processed, embedded in paraffin, and sectioned to 4 µm thickness. The tissue sections were then stained using the GMS stain (Grocott Methenamine Silver stain) as per the manufacturer's protocol (Cat. #AB287884). All the experiments were performed independently, thrice.

### Illustrations

Figure illustrations were created using Biorender (https://app.biorender.com).

## Acknowledgements

We thank Dr. Rajan Sankaranarayanan (CSIR-CCMB) and Dr. Sunil Laxman (BRIC-inStem) for critical reading of our manuscript. We gratefully acknowledge the invaluable support and resources offered by CSIR-Centre for Cellular and Molecular Biology (CCMB) central facilities, including Advanced Microscopy and Imaging Facility (AMIF), Next Generation Sequencing (NGS) Facility, Tissue Culture Facility, and Animal House Facility. We acknowledge help from T Avinash Raj (CSIR- CCMB) for

preparation of samples for histological analysis. DS thanks the University Grants Commission (UGC), India, for their research fellowship. SV acknowledges funding from the Indian Council of Medical Research (ICMR) (IIRPSG-2024-01-02717), India; Anusandhan National Research Foundation (ANRF) (SRG/2023/000470), India; Council of Scientific & Industrial Research (CSIR) (FTT070505), India; and DBT-Wellcome Trust India Alliance (IA/E/16/1/502996), India.

# Additional information

## Funding

| Funder | Grant reference number | Author |
|---|---|---|
| Indian Council of Medical Research | IIRPSG-2024-01-02717 | Sriram Varahan |
| Anusandhan National Research Foundation | SRG/2023/000470 | Sriram Varahan |
| Council of Scientific and Industrial Research | FTT070505 | Sriram Varahan |
| DBT-Wellcome Trust India Alliance | IA/E/16/1/502996 | Sriram Varahan |

The funders had no role in study design, data collection and interpretation, or the decision to submit the work for publication. For the purpose of Open Access, the authors have applied a CC BY public copyright license to any Author Accepted Manuscript version arising from this submission.

## Author contributions

Dhrumi Shah, Conceptualization, Formal analysis, Investigation, Visualization, Writing – original draft; Nikita Rewatkar, Adishree M, Siddhi Gupta, Sudharsan Mathivathanan, Sayantani Biswas, Investigation; Sriram Varahan, Conceptualization, Formal analysis, Supervision, Funding acquisition, Investigation, Visualization, Writing – original draft, Writing – review and editing

## Author ORCIDs

Dhrumi Shah  https://orcid.org/0009-0005-9108-3933
Sriram Varahan  https://orcid.org/0000-0002-3609-4032

## Ethics

All animal experiments in this manuscript were reviewed and approved by the Institutional Animal Ethics Committee (IAEC) of CSIR-Centre for Cellular and Molecular Biology (Approval number: 52/2024-C).

Reviewer #1 (Public review): https://doi.org/10.7554/eLife.109075.3.sa1
Reviewer #2 (Public review): https://doi.org/10.7554/eLife.109075.3.sa2
Reviewer #3 (Public review): https://doi.org/10.7554/eLife.109075.3.sa3
Author response https://doi.org/10.7554/eLife.109075.3.sa4

# Additional files

## Supplementary files

MDAR checklist

Supplementary file 1. Yeast strains used in this study.

Supplementary file 2. Oligonucleotides used in this sUsed in this Study.

## Data availability

All relevant data are within the paper and its supporting information files, except for RNA-Seq data and the Whole Genome Sequencing data. RNA-Seq data and the Whole Genome Sequencing data

are deposited in NCBI's SRA database and are accessible through BioProject accession number PRJNA1263201. (https://www.ncbi.nlm.nih.gov/bioproject/PRJNA1263201).

The following dataset was generated:

| Author(s) | Year | Dataset title | Dataset URL | Database and Identifier |
|---|---|---|---|---|
| Shah et al. | 2025 | Glycolysis-dependent Sulfur Metabolism Orchestrates Morphological Plasticity and Virulence in Fungi (RNA-Seq and Whole Genome Sequencing) | https://www.ncbi.nlm.nih.gov/bioproject/PRJNA1263201 | NCBI BioProject, PRJNA1263201 |

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
