## [Editor Report · eLife Assessment]

This work identifies a novel, conserved link between glycolysis and sulfur metabolism that governs fungal morphogenesis and virulence. The **compelling** evidence, integrating multiple approaches, provides an **important** conceptual advance. A future mechanistic dissection of how sulfur metabolites interface with known pathways is encouraged.

---

## [Referee Report · Reviewer #1 (Public review)]

Summary:

Fungal survival and pathogenicity rely on the ability to undergo reversible morphological transitions, which is often linked to nutrient availability. In this study, the authors uncover a conserved connection between glycolytic activity and sulfur amino acid biosynthesis that drives morphogenesis in two fungal model systems. By disentangling this process from canonical cAMP signaling, the authors identify a new metabolic axis that integrates central carbon metabolism with developmental plasticity and virulence.

Strengths:

The study integrates different experimental approaches, including genetic, biochemical, transcriptomic and morphological analyses and convincingly demonstrates that perturbations in glycolysis alters sulfur metabolic pathways and thus impacts pseudohyphal and hyphal differentiation. Overall, this work offers new and important insights into how metabolic fluxes are intertwined with fungal developmental programs and therefore opens new perspectives to investigate morphological transitioning in fungi.

Importantly, in the revised version the authors now substantiate the transcriptomic findings by RT-qPCR analyses in the pfk1ΔΔ and adh1ΔΔ strains, demonstrating that genetic disruption of glycolytic flux generally mirrors the effects of 2-deoxyglucose treatment. The manuscript's discussion has also been strengthened by explicitly addressing why cysteine and methionine differ in their ability to rescue filamentation in *S. cerevisiae* versus *C. albicans*, highlighting species-specific differences in sulfur uptake and transsulfuration pathways.

Overall, this revised manuscript provides compelling evidence for a previously unrecognized coupling between glycolysis and sulfur metabolism that shapes fungal morphogenesis and virulence. It opens new perspectives on metabolic control of fungal development and raises interesting mechanistic questions for future work.

Comments on revisions:

The authors have incorporated all of my suggested changes and addressed all raised concerns.

---

## [Referee Report · Reviewer #2 (Public review)]

Summary:

The manuscript investigates the interplay between glycolysis and sulfur metabolism in regulating fungal morphogenesis and virulence. Using both *Saccharomyces cerevisiae* and Candida albicans, the authors demonstrate that glycolytic flux is essential for morphogenesis under nitrogen-limiting conditions, acting independently of the established cAMP-PKA pathway. Transcriptomic and genetic analyses reveal that glycolysis influences the de novo biosynthesis of sulfur-containing amino acids, specifically cysteine and methionine. Notably, supplementation with sulfur sources restores morphogenetic and virulence defects in glycolysis-deficient mutants, thereby linking core carbon metabolism with sulfur assimilation and fungal pathogenicity.

Strengths:

The work identifies a previously uncharacterized link between glycolysis and sulfur metabolism in fungi, bridging metabolic and morphogenetic regulation which is an important conceptual advance and fungal pathogenicity. Demonstrating that adding cysteine supplementation rescues virulence defects in animal model connects basic metabolism to infection outcomes that add on biomedical importance.

Comments on revisions:

The authors have sufficiently addressed my concern and provided a clear justification for their proposed model including the limitations of performing the mechanistic assays at this stage. I am satisfied with the response and have no further comments

---

## [Referee Report · Reviewer #3 (Public review)]

This study investigates the connection between glycolysis and the biosynthesis of sulfur-containing amino acids in controlling fungal morphogenesis, using *Saccharomyces cerevisiae* and C. albicans as model organisms. The authors identify a conserved metabolic axis that integrates glycolysis with cysteine/methionine biosynthetic pathways to influence morphological transitions. This work broadens the current understanding of fungal morphogenesis, which has largely focused on gene regulatory networks and cAMP-dependent signaling pathways, by emphasizing the contribution of metabolic control mechanisms.

Strengths:

The delineation of how glycolytic flux regulates fungal morphogenesis through a cAMP-independent mechanism is an advancement. The coupling of glycolysis with the de novo biosynthesis of sulfur-containing amino acids, a requirement for morphogenesis, introduces a novel and unexpected layer of regulation.

Demonstrating this mechanism in both *S. cerevisiae* and *C. albicans* strengthens the argument for its evolutionary conservation and biological importance.

The ability to rescue the morphogenesis defect through supplementation of sulfur-containing amino acids provides a functional validation.

Weaknesses:

cAMP addition rescued the pseudohyphal differentiation defect exhibited by the ΔΔgpa2 strain. More clarity is needed on how this mechanism is mechanistically distinct from the metabolic control - whether cAMP acts in parallel or downstream to sulfur-containing amino acids biosynthesis has to be characterized. Supplementation of cysteine and methionine bypasses glycolytic regulation; the link between these amino acids and their role in fungal morphogenesis is not completely characterized.

The demonstrated link between glycolysis and sulfur amino acid biosynthesis, along with its implications for virulence in C. albicans, is important for understanding fungal adaptation, as mentioned in the article; however, the downstream effects of Met4 activation were not fully characterized. How does Cysteine/Methionine rescue morphogenesis? The author's response figure 1 shows that there are no significant transcriptional changes in the expression of cAMP-PKA pathway-associated genes, which alone could not completely explain the role of gpa2 in morphogenesis, because exogenous cAMP can restore pseudohyphal differentiation in the ΔΔgpa2 background (Revised Fig. 1L). This implies that gpa2's function in morphogenesis is an additional, or possibly a metabolic or post-transcriptional, layer of regulation, and its connection to sulfur-containing amino acids remains to be elucidated.

---

## [Author Response]

The following is the authors’ response to the original reviews.

**Public Reviews:**

**Reviewer #1 (Public review):**
Summary:Fungal survival and pathogenicity rely on the ability to undergo reversible morphological transitions, which are often linked to nutrient availability. In this study, the authors uncover a conserved connection between glycolytic activity and sulfur amino acid biosynthesis that drives morphogenesis in two fungal model systems. By disentangling this process from canonical cAMP signaling, the authors identify a new metabolic axis that integrates central carbon metabolism with developmental plasticity and virulence.Strengths:The study integrates different experimental approaches, including genetic, biochemical, transcriptomic, and morphological analyses, and convincingly demonstrates that perturbations in glycolysis alter sulfur metabolic pathways and thus impact pseudohyphal and hyphal differentiation. Overall, this work offers new and important insights into how metabolic fluxes are intertwined with fungal developmental programs and therefore opens new perspectives to investigate morphological transitioning in fungi.

We thank the reviewer for finding this study to be of importance and for appreciating our multipronged approach to substantiate our finding that perturbations in glycolysis alter sulfur metabolism and thus impact pseudohyphal and hyphal differentiation in fungi.

Weaknesses:A few aspects could be improved to strengthen the conclusions. Firstly, the striking transcriptomic changes observed upon 2DG treatment should be analyzed in *S. cerevisiae* adh1 and pfk1 deletion strains, for instance, through qPCR or western blot analyses of sulfur metabolism genes, to confirm that observed changes in 2DG conditions mirror those seen in genetic mutants. Secondly, differences between methionine and cysteine in their ability to rescue the mutant phenotype in both species are not mentioned, nor discussed in more detail. This is especially important as there seem to be differences between *S. cerevisiae* and *C. albicans*, which might point to subtle but specific metabolic adaptations.The authors are also encouraged to refine several figure elements for clarity and comparability (e.g., harmonized axes in bar plots), condense the discussion to emphasize the conceptual advances over a summary of the results, and shorten figure legends.

We are grateful for this valuable and constructive feedback, and we agree with the reviewer on the necessity of performing RT-qPCR analysis of sulfur metabolism genes in ∆∆pfk1 and ∆∆adh1 strains of *S. cerevisiae* to validate our RNA-Seq results using 2DG. We have performed this experiment, and our results show that several genes involved in the de novo biosynthesis of sulfur-containing amino acids are downregulated in both the ∆∆pfk1 and ∆∆adh1 strains, corroborating the downregulation of sulfur metabolism genes in the 2DG treated samples. This new data is now included in the revised manuscript as Supplementary Figure 2C.

Furthermore, we acknowledge the reviewer’s point regarding the significance of comparing the differences in the ability of methionine and cysteine to rescue filamentation defects exhibited by the mutants, between *S. cerevisiae* and *C. albicans*. The observed differences between *S. cerevisiae* and C. albicans likely highlight species-specific metabolic adaptations within the sulfur assimilation pathway. While both yeasts employ the transsulfuration pathway to interconvert these sulfur-containing amino acids, the precise regulatory points including the specific enzymes, their compartmentalization, and transcriptional control are not identical. For instance, differences in the feedback inhibition mechanisms or the expression levels of key transsulfuration enzymes between *S. cerevisiae* and C. albicans could explain the variations in the phenotypic rescue experiments (Chebaro et al., 2017; Lombardi et al., 2024; Rouillon et al., 2000; Shrivastava et al., 2021; Thomas and Surdin-Kerjan, 1997). Furthermore, the species-specific differences in amino acid transport systems (permeases) adds another layer of complexity. *S. cerevisiae* primarily uses multiple, low-affinity permeases for cysteine transport (Gap1, Bap2, Bap3, Tat1, Tat2, Agp1, Gnp1, Yct1), while relying on a limited set of high-affinity transporters (like Mup1) for methionine transport, with the added complexity that its methionine transporters can also transport cysteine (Düring-Olsen et al., 1999; Huang et al., 2017; Kosugi et al., 2001; Menant et al., 2006). In contrast, *C. albicans* utilizes a high-affinity transporters for the uptake of both amino acids, employing Cyn1 specifically for cysteine and Mup1 for methionine, indicating a greater reliance on dedicated transport mechanisms for these sulfur-containing molecules in the pathogenic yeast (Schrevens et al., 2018; Yadav and Bachhawat, 2011). A combination of the aforesaid factors could be the potential reason for the differences in the ability of cysteine and methionine to rescue filamentation in *S. cerevisiae* and *C. albicans*.

Finally, we have enhanced the quantitative rigor and clarity of the data presentation in the revised manuscript by implementing Y-axis uniformity across all relevant bar graphs to facilitate a more robust and direct comparative analysis. We have also condensed the discussion to emphasize the conceptual advances and have shortened the figure legends as per the reviewer suggestions

**Reviewer #2 (Public review):**
Summary:This manuscript investigates the interplay between glycolysis and sulfur metabolism in regulating fungal morphogenesis and virulence. Using both *Saccharomyces cerevisiae* and Candida albicans, the authors demonstrate that glycolytic flux is essential for morphogenesis under nitrogen-limiting conditions, acting independently of the established cAMP-PKA pathway. Transcriptomic and genetic analyses reveal that glycolysis influences the de novo biosynthesis of sulfur-containing amino acids, specifically cysteine and methionine. Notably, supplementation with sulfur sources restores morphogenetic and virulence defects in glycolysis-deficient mutants, thereby linking core carbon metabolism with sulfur assimilation and fungal pathogenicity.Strengths:The work identifies a previously uncharacterized link between glycolysis and sulfur metabolism in fungi, bridging metabolic and morphogenetic regulation, which is an important conceptual advance and fungal pathogenicity. Demonstrating that adding cysteine supplementation rescues virulence defects in animal models connects basic metabolism to infection outcomes, which adds to biomedical importance.

We would like to thank the reviewer for the positive comments on our work. We are pleased that they recognize the novel metabolic link between glycolysis and sulfur metabolism as a key conceptual advance in fungal morphogenesis.

Weaknesses:The proposed model that glycolytic flux modulates Met30 activity post-translationally remains speculative. While data support Met4 stabilization in met30 deletion strains, the mechanism of Met30 modulation by glycolysis is not demonstrated.

We thank the reviewer for this valuable feedback. The activity of the SCF^Met30^ E3 ubiquitin ligase, mediated by the F box protein Met30, is dynamically regulated through both proteolytic degradation and its dissociation from the SCF complex, to coordinate sulfur metabolism and cell cycle progression (Smothers et al., 2000; Yen et al., 2005). Our transcriptomic (RNA-seq analysis) and protein expression analysis (Fig. 3J) confirms that Met30 expression is not differentially regulated in the presence of 2DG, effectively eliminating changes in synthesis or SCF^Met30^ proteasomal degradation as the dominant regulatory mechanism. This observation is consistent with the established paradigm wherein stress signals, such as cadmium (Cd^2+^) exposure, rapidly inactivates the SCF^Met30^ E3 ubiquitin ligase via the dissociation of Met30 from the Skp1 subunit of the SCF complex (Lauinger et al., 2024; Yen et al., 2005). We therefore propose that active glycolytic flux modulates SCF^Met30^ activity post-translationally, specifically by triggering Met30 detachment from the SCF complex. This mechanism would stabilize the primary substrate, the transcription factor Met4, thus promoting the biosynthesis of sulfur-containing amino acids. Mechanistic validation of this hypothesis, particularly the assessment of Met30 dissociation from the SCF^Met30^ complex via immunoprecipitation (IP), is technically challenging. Since these experiments will involve isolation of cells from colonies undergoing pseudohyphal differentiation, on solid media (given that pseudohyphal differentiation does not occur in liquid media that is limiting for nitrogen (Gancedo, 2001; Gimeno et al., 1992)), current cell yields (OD_600_≈1 from ≈80-100 colonies) are significantly below the amount of cells that is needed to obtain the required amount of total protein concentration, for standard pull down assays (OD_600_≈600-800 is required to achieve 1-2 mg/ml of total protein which is the standard requirement for pull down protocols in *S. cerevisiae* (Lauinger et al., 2024)).

Given that the primary objective of our study is to establish the novel regulatory link between glycolysis and sulfur metabolism in the context of fungal morphogenesis, we would like to explore these crucial mechanistic details, in depth, in a subsequent study.

**Reviewer #3 (Public review):**
This study investigates the connection between glycolysis and the biosynthesis of sulfur-containing amino acids in controlling fungal morphogenesis, using *Saccharomyces cerevisiae* and C. albicans as model organisms. The authors identify a conserved metabolic axis that integrates glycolysis with cysteine/methionine biosynthetic pathways to influence morphological transitions. This work broadens the current understanding of fungal morphogenesis, which has largely focused on gene regulatory networks and cAMP-dependent signaling pathways, by emphasizing the contribution of metabolic control mechanisms. However, despite the novel conceptual framework, the study provides limited mechanistic characterization of how the sulfur metabolism and glycolysis blockade directly drive morphological outcomes. In particular, the rationale for selecting specific gene deletions, such as Met32 (and not Met4), or the Met30 deletion used to probe this pathway, is not clearly explained, making it difficult to assess whether these targets comprehensively represent the metabolic nodes proposed to be critical. Further supportive data and experimental validation would strengthen the claims on connections between glycolysis, sulfur amino acid metabolism, and virulence.Strengths:(1) The delineation of how glycolytic flux regulates fungal morphogenesis through a cAMP-independent mechanism is a significant advancement. The coupling of glycolysis with the de novo biosynthesis of sulfur-containing amino acids, a requirement for morphogenesis, introduces a novel and unexpected layer of regulation.(2) Demonstrating this mechanism in both *S. cerevisiae* and *C. albicans* strengthens the argument for its evolutionary conservation and biological importance.(3) The ability to rescue the morphogenesis defect through exogenous supplementation of sulfur-containing amino acids provides functional validation.(4) The findings from the murine Pfk1-deficient model underscore the clinical significance of metabolic pathways in fungal infections.

We are grateful for this comprehensive and insightful summary of our work. We deeply appreciate the reviewer's recognition of the key conceptual breakthroughs regarding the metabolic regulation of fungal morphogenesis and the clinical relevance of our findings.

Weaknesses:(1) While the link between glycolysis and sulfur amino acid biosynthesis is established via transcriptomic and proteomic analysis, the specific regulation connecting these pathways via Met30 remains to be elucidated. For example, what are the expression and protein levels of Met30 in the initial analysis from Figure 2? How specific is this effect on Met30 in anaerobic versus aerobic glycolysis, especially when the pentose phosphate pathway is involved in the growth of the cells when glycolysis is perturbed ?

We are grateful for the insightful feedback provided by the reviewer. *S. cerevisiae* is a Crabtree positive organism that primarily uses anaerobic glycolysis to metabolize glucose, under glucose-replete conditions (Barford and Hall, 1979; De Deken, 1966) and our pseudohyphal differentiation assays are performed in glucose-rich conditions (Gimeno et al., 1992). Furthermore, perturbation of glycolysis is known to induce compensatory upregulation of the Pentose Phosphate Pathway (PPP) (Ralser et al., 2007) and we have also observed the upregulation of the gene that encodes for transketolase-1 (Tkl1), a key enzyme in the PPP, in our RNA-seq data. Importantly, our transcriptomic (RNA-seq analysis) and protein expression analysis (Fig. 3J) confirms that Met30 expression is not differentially regulated in the presence of 2DG, effectively eliminating changes in synthesis or SCF^Met30^ proteasomal degradation as the dominant regulatory mechanism. This aligns with the established paradigm wherein stress signals, such as cadmium (Cd^2+^) exposure, rapidly inactivates SCF^Met30^ E3 ubiquitin ligase via Met30 dissociation from the Skp1 subunit of the complex (Lauinger et al., 2024; Yen et al., 2005). We therefore propose that active glycolytic flux modulates SCF^Met30^ activity post-translationally, specifically by triggering Met30 detachment from the SCF complex. This mechanism would stabilize the primary substrate, the transcription factor Met4, thus promoting the biosynthesis of sulfur-containing amino acids. Further experiments are required to delineate the specific role of pentose phosphate pathway in the aforesaid proposed regulation of the Met30 activity under glycolysis perturbation and this will be explored in our subsequent study.

(2) Including detailed metabolite profiling could have strengthened the metabolic connection and provided additional insights into intermediate flux changes, i.e., measuring levels of metabolites to check if cysteine or methionine levels are influenced intracellularly. Also, it is expected to see how Met30 deletion could affect cell growth. Data on Met30 deletion and its effect on growth are not included, especially given that a viable heterozygous Met30 strain has been established. Measuring the cysteine or methionine levels using metabolomic analysis would further strengthen the claims in every section.

We are grateful to the reviewer for this constructive feedback. To address the potential impact of met30 deletion on cell growth, we have included new data (Suppl. Fig. 4A) demonstrating that the deletion of a single copy of met30 in diploid *S. cerevisiae* does not compromise overall cell growth under nitrogen-limiting conditions as the ∆met30 strain grows similar to the wild-type strain.

Our pseudohyphal/hyphal differentiation assays show that the defects induced by glycolytic perturbation is fully rescued by the exogenous supplementation of sulfur-containing amino acids, cysteine or methionine. Since these data conclusively demonstrate that the primary metabolic limitation caused by the perturbation of glycolysis, which leads to filamentation defects is sulfur metabolism, we posit that performing comprehensive metabolic profiling would primarily reconfirm the aforesaid results. We believe that our in vitro and in vivo sulfur add-back experiments sufficiently substantiate the novel regulatory metabolic link between glycolysis and sulfur metabolism.

(3) In comparison with the previous bioRxiv (doi: https://doi.org/10.1101/2025.05.14.654021) of this article in May 2025 to the recent bioRxiv of this article (doi: https://doi.org/10.1101/2025.05.14.654021), there have been some changes, and Met30 deletion has been recently included, and the chemical perturbation of glycolysis has been added as new data. Although the changes incorporated in the recent version of the article improved the illustration of the hypothesis in Figure 6, which connects glycolysis to Sulfur metabolism, the gene expression and protein levels of all genes involved in the illustrated hypothesis are not consistently shown. For example, in some cases, the Met4 expression is not shown (Figure 4), and the Met30 expression is not shown during profiling (gene expression or protein levels) throughout the manuscript. Lack of consistency in profiling the same set of key genes makes understanding more complicated.

We thank the reviewer for this feedback which helps us to clarify the scope of our transcriptomic analysis. Our decision to focus our RT-qPCR experiments on downstream targets, while excluding met4 and met30 from the RT-qPCR analysis, is based on their known regulatory mechanisms. Met4 activity is predominantly regulated by post-translational ubiquitination by the SCFMet30 complex followed by its degradation (Rouillon et al., 2000; Shrivastava et al., 2021; Smothers et al., 2000) while Met30 activity is primarily regulated by its auto-degradation or its dissociation from the SCFMet30 complex (Lauinger et al., 2024; Smothers et al., 2000; Yen et al., 2005). Consistent with this, our RNA-Seq results indicate that neither met4 nor met30 transcripts are differentially expressed, in response to 2DG addition. For all our RT-qPCR analysis in *S. cerevisiae* and *C. albicans*, we have consistently used the same set of sulfur metabolism genes and these include met32, met3, met5, met10 and met17. Our data on protein expression analysis of Met30 in *S. cerevisiae* (Fig. 3J) confirms that Met30 expression is not differentially regulated in the presence of 2DG, effectively eliminating changes in synthesis or SCFMet30 proteasomal degradation as the dominant regulatory mechanism.

(4) The demonstrated link between glycolysis and sulfur amino acid biosynthesis, along with its implications for virulence in C. albicans, is important for understanding fungal adaptation, as mentioned in the article; however, the Met4 activation was not fully characterized, nor were the data presented when virulence was assessed in Figure 4. Why is Met4 not included in Figure 4D and I? Especially, according to Figure 6, Met4 activation is crucial and guides the differences between glycolysis-active and inactive conditions.

We thank the reviewer for their input. As the Met4 transcription factor in C. albicans is primarily regulated post-translationally through its degradation and inactivation by the SCFMet30 E3 ubiquitin ligase complex (Shrivastava et al., 2021), we opted to monitor the transcriptional status of downstream targets of Met4 (i.e., genes directly regulated by Met4), as these are the genes that exhibit the most direct and functionally relevant transcriptional changes in response to the altered Met4 levels.

(5) Similarly, the rationale behind selecting Met32 for characterizing sulfur metabolism is unclear. Deletion of Met32 resulted in a significant reduction in pseudohyphal differentiation; why is this attributed only to Met32? What happens if Met4 is deleted? It is not justified why Met32, rather than Met4, was chosen. Figure 6 clearly hypothesizes that Met4 activation is the key to the mechanism.

We sincerely thank the reviewer for this insightful query regarding our selection of the met32 for our gene deletion experiments. The choice of ∆∆met32 strain was strategically motivated by its unique phenotypic properties within the de novo biosynthesis of sulfur-containing amino acids pathway. While deletions of most the genes that encode for proteins involved in the de novo biosynthesis of sulfurcontaining amino acids, result in auxotrophy for methionine or cysteine, ∆∆met32 strain does not exhibit this phenotype (Blaiseau et al., 1997). This key distinction is attributed to the functional redundancy provided by the paralogous gene, met31 (Blaiseau et al., 1997). Crucially, given that the deletion of the central transcriptional regulator, met4, results in cysteine/methionine auxotrophy, the use of the ∆∆met32 strain provides an essential, viable experimental model for investigating the role of sulfur metabolism during pseudohyphal differentiation in *S. cerevisiae*.

(6) The comparative RT-qPCR in Figure 5 did not account for sulfur metabolism genes, whereas it was focused only on virulence and hyphal differentiation. Is there data to support the levels of sulfur metabolism genes?

We thank the reviewer for this feedback. We wish to respectfully clarify that the data pertaining to expression of sulfur metabolism genes in the presence of 2DG or in the ∆∆pfk1 strain in C. albicans are already included and discussed within the manuscript. These results can be found in Figure 4, panels D and I, respectively.

(7) To validate the proposed interlink between sulfur metabolism and virulence, it is recommended that the gene sets (illustrated in Figure 6) be consistently included across all comparative data included throughout the comparisons. Excluding sulfur metabolism genes in Figure 5 prevents the experiment from demonstrating the coordinated role of glycolysis perturbation → sulfur metabolism → virulence. The same is true for other comparisons, where the lack of data on Met30, Met4, etc., makes it hard.to connect the hypothesis. It is also recommended to check the gene expression of other genes related to the cAMP pathway and report them to confirm the cAMP-independent mechanism. For example, gap2 deletion was used to confirm the effects of cAMP supplementation, but the expression of this gene was not assessed in the RNA-seq analysis in Figure 2. It would be beneficial to show the expression of cAMP-related genes to completely confirm that they do not play a role in the claims in Figure 2.

We thank the reviewer for this valuable feedback. The transcriptional analysis of the sulfur metabolism genes in the presence of 2DG and the ∆∆pfk1 strain is shown in Figures 4D and 4I.

Our RNA-seq analysis (Author response image 1) confirms that there is no significant transcriptional change in the expression of cAMP-PKA pathway associated genes (Log2 fold change ≥ 1 for upregulated genes and Log2 fold change ≤ -1 for downregulated genes) in 2DG treated cells compared to the untreated control cells, reinforcing our conclusion that the glycolytic regulation of fungal morphogenesis is mediated through a cAMP-PKA-independent mechanism.

**Author response image 1. sa4fig1:** 

(8) Although the NAC supplementation study is included in the new version of the article compared to the previous version in BioRxiv (May 2025), the link to sulfur metabolism is not well characterized in Figure 5 and their related datasets. The main focus of the manuscript is to delineate the role of sulfur metabolism; hence, it is anticipated that Figure 5 will include sulfur-related metabolic genes and their links to pfk1 deletion, using RT-PCR measurements as shown for the virulence genes.

We thank the reviewer for this question. The relevant data are indeed present within the current submission. We respectfully direct the reviewer's attention to Figure 4, panels D and I, where the data pertaining to expression of sulfur metabolism genes in the presence of 2DG or in the ∆∆pfk1 strain in C. albicans can be found.

(9) The manuscript would benefit from more information added to the introduction section and literature supports for some of the findings reported earlier, including the role of (i) cAMP-PKA and MAPK pathways, (ii) what is known in the literature that reports about the treatment with 2DG (role of Snf1, HXT1, and HXT3), as well as how gpa2 is involved. Some sentences in the manuscripts are repetitive; it would be beneficial to add more relevant sections to the introduction and discussion to clarify the rationale for gene choices.

We thank the reviewer for this valuable feedback. We have incorporated these changes in our revised manuscript.

**Recommendations for the authors:**

**Reviewer #1 (Recommendations for the authors):**
(1) Line 107: As morphological transitions are indeed a conserved phenomenon across fungal species, hosts & environmental niches, the authors could refer to a few more here (infection structures like appressoria; fruiting bodies, etc.).

We thank the reviewer for this valuable feedback. We have incorporated these changes in our revised manuscript.

Line 119/120: That's a bit misleading in my opinion. Gpr1 acts as a key sensor of external carbon, while Ras proteins control the cAMP pathway as intracellular sensory proteins. That should be stated more clearly. cAMP is the output and not the sensor.

We appreciate the reviewer's detailed attention to this signaling network. We have revised the manuscript to precisely reflect this established signaling hierarchy for maximum clarity.

(2) Line 180: ..differentiation

We thank the reviewer for this valuable feedback. We have incorporated this change in our revised manuscript.

(3) Figure 1 panels C & F. The authors should provide the same scale for all experiments. Otherwise, the interpretation can be difficult. The same applies to the different bar plots in Figure 4. Have the authors quantified pseudohyphal differentiation in the cAMP add-back assays? I agree that the chosen images look convincing, but they don't reflect quantitative analyses.

We thank the reviewer for detailed and constructive feedback. We have changed the Y-axis and made it more uniform to improve the clarity of our data presentation in the revised manuscript.

We have also incorporated the quantitative analysis of the cAMP add-back assays in *S. cerevisiae*, in Figure 2 Panel L.

(4) Line 367/68: "cysteine or methionine was able to completely rescue". Here, the authors should phrase their wording more carefully. Figure 3C shows the complete rescue of the phenotype qualitatively, but Figure 3D clearly shows that there are differences between the supplementation of cysteine and methionine, with the latter not fully restoring the phenotype.

We sincerely appreciate the reviewer's meticulous attention to the data interpretation. We fully agree that the initial phrasing in lines 367/368 requires adjustment, as Figure 3D establishes a quantitative difference in the efficiency of phenotypic rescue between cysteine and methionine supplementation. We have revised the text to articulate this difference.

(5) Line 568: Here, apparently, the ability to rescue the differentiation phenotype is reversed compared to the experiment with *S. cerevisiae*. Cysteine only results in ~20% hyphal cells, while methionine restores to wild-type-like hyphal formation. Can the authors comment on where these differences might originate from? Is there a difference in the uptake of cysteine vs. methionine in the two species or consumption rates?

We thank the reviewer for their detailed and constructive feedback. We believe this phenotypic difference can be due to the distinct metabolic prioritization of sulfur amino acids in C. albicans. Methionine is a known trigger for hyphal differentiation in C. albicans and serves as the immediate precursor for the universal methyl donor, S-adenosylmethionine (SAM) (Schrevens et al., 2018). (Kraidlova et al., 2016). The morphological transition to hyphae involves a complex regulatory cascade which requires high rates of methylation, and this requires a rapid and direct conversion of methionine into SAM (Kraidlova et al., 2016; Schrevens et al., 2018). Cysteine, however, must first be converted into methionine via the transsulfuration pathway to produce SAM, making it metabolically less efficient for these aforesaid processes.

**Reviewer #2 (Recommendations for the authors):**

The study's comprehensive experimental approach with integrating pharmacological inhibition, genetic manipulation, transcriptomics, and infection animal model, provides strong evidence for a conserved mechanism, though some aspects need further clarification.

Major Comments:(1) While the data suggest that glycolysis affects Met30 activity post-translationally, the underlying mechanism remains speculative. The authors should perform co-immunoprecipitation or ubiquitination assays to confirm whether glycolytic perturbation alters Met30-SCF complex interactions or Met4 ubiquitination levels.

We thank the reviewer for this valuable feedback. The activity of the SCF^Met30^ E3 ubiquitin ligase, mediated by the F box protein Met30, is dynamically regulated through both proteolytic degradation and its dissociation from the SCF complex, to coordinate sulfur metabolism and cell cycle progression (Smothers et al., 2000; Yen et al., 2005). Our transcriptomic (RNA-seq analysis) and protein expression analysis (Fig. 3J) confirms that Met30 expression is not differentially regulated in the presence of 2DG, effectively eliminating changes in synthesis or SCF^Met30^ proteasomal degradation as the dominant regulatory mechanism. This observation is consistent with the established paradigm wherein stress signals, such as cadmium (Cd^2+^) exposure, rapidly inactivates the SCF^Met30^ E3 ubiquitin ligase via the dissociation of Met30 from the Skp1 subunit of the SCF complex (Lauinger et al., 2024; Yen et al., 2005). We therefore propose that active glycolytic flux modulates SCF^Met30^ activity post-translationally, specifically by triggering Met30 detachment from the SCF complex. This mechanism would stabilize the primary substrate, the transcription factor Met4, thus promoting the biosynthesis of sulfur-containing amino acids. Mechanistic validation of this hypothesis, particularly the assessment of Met30 dissociation from the SCF^Met30^ complex via immunoprecipitation (IP), is technically challenging. Since these experiments will involve isolation of cells from colonies undergoing pseudohyphal differentiation, on solid media (given that pseudohyphal differentiation does not occur in liquid media that is limiting for nitrogen (Gancedo, 2001; Gimeno et al., 1992)), current cell yields (OD^600^≈1 from ≈80-100 colonies) are significantly below the amount of cells that is needed to obtain the required amount of total protein concentration, for standard pull down assays (OD600≈600-800 is required to achieve 1-2 mg/ml of total protein which is the standard requirement for pull down protocols in *S. cerevisiae* (Lauinger et al., 2024)).

Given that the primary objective of our study is to establish the novel regulatory link between glycolysis and sulfur metabolism in the context of fungal morphogenesis, we would like to explore these crucial mechanistic details, in depth, in a subsequent study.

(2) 2DG can exert pleiotropic effects unrelated to glycolytic inhibition (e.g., ER stress, autophagy induction). The authors are encouraged to perform complementary metabolic flux analyses, such as quantification of glycolytic intermediates or ATP levels, to confirm specific glycolytic inhibition.

We appreciate the reviewer's concern regarding the potential pleiotropic effects of 2DG. While we acknowledge that 2DG may induce secondary cellular stress, we are confident that the observed phenotypes are robustly attributed to glycolytic inhibition based on our complementary genetic evidence. Specifically, the deletion strains ∆∆pfk1 and ∆∆adh1, which genetically perturb distinct steps in glycolysis, recapitulate the phenotypic results observed with 2DG treatment. Given this strong congruence between chemical inhibition and specific genetic deletions of key glycolytic enzymes, we are confident that our observed phenotypes are predominantly driven by the perturbation of the glycolytic pathway by 2DG.

(3) The differential rescue effects (cysteine-only in inhibitor assays vs. both cysteine and methionine in genetic mutants) require further explanation. The authors should discuss potential differences in metabolic interconversion or amino acid transport that may account for this observation.

We thank the reviewer for their valuable feedback. One explanation for the observed differential rescue effects of cysteine and methionine can be due to the distinct amino acid transport systems used by *S. cerevisiae* to transport these amino acids. *S. cerevisiae* primarily uses multiple, lowaffinity permeases (Gap1, Bap2, Bap3, Tat1, Tat2, Agp1, Gnp1, Yct1) for cysteine transport, while relying on a limited set of high-affinity transporters (like Mup1) for methionine transport, with the added complexity that its methionine transporters can also transport cysteine (Düring-Olsen et al., 1999; Huang et al., 2017; Kosugi et al., 2001; Menant et al., 2006). Hence, it is likely that cysteine uptake could be happening at a higher efficiency in *S. cerevisiae* compared to methionine uptake. Therefore, to achieve a comparable functional rescue by exogenous supplementation of methionine, it is necessary to use a higher concentration of methionine. When we performed our rescue experiments using higher concentrations of methionine, we did not see any rescue of pseudohyphal differentiation in the presence of 2DG and in fact we noticed that, at higher concentrations of methionine, the wild-type strain failed to undergo pseudohyphal differentiation even in the absence of 2DG. This is likely due to the fact that increasing the methionine concentration raises the overall nitrogen content of the medium, thereby making the medium less nitrogen-starved. This presents a major experimental constraint, as pseudohyphal differentiation is strictly dependent on nitrogen limitation, and the elevated nitrogen resulting from the higher methionine concentration can inhibit pseudohyphal differentiation.

(4) NAC may influence host redox balance or immune responses. The discussion should consider whether the observed virulence rescue could partly result from host-directed effects.

We thank the reviewer for this valuable feedback. We acknowledge the role of NAC in host directed immune response. It is important to note that, in the context of certain bacterial pathogens, NAC has been reported to augment cellular respiration, subsequently increasing Reactive Oxygen Species (ROS) generation, which contributes to pathogen clearance (Shee et al., 2022). Interestingly, in our study, NAC supplementation to the mice was given prior to the infection and maintained continuously throughout the duration of the experiment. This continuous supply of NAC likely contributes to the rescue of virulence defects exhibited by the ∆∆pfk1 strain (Fig. 5I and J). Essentially, NAC likely allows the mutant to fully activate its essential virulence strategies (including morphological switching), to cause a successful infection in the host. As per the reviewer suggestion, this has been included in the discussion section of the manuscript.

**Reviewer #3 (Recommendations for the authors):**
Most of the comments related to improving the manuscript have been provided in the public review. Here are some specifics for the authors to consider:(1) It is important to clarify the rationale for choosing specific gene deletions over other key genes (e.g., Met32 and Met30) and explain why Met4 was not included, given its proposed central role in Figure 6.

We sincerely thank the reviewer for this insightful query regarding our selection of the met32 for our gene deletion experiments. The choice of ∆∆met32 strain was strategically motivated by its unique phenotypic properties within the de novo biosynthesis of sulfur-containing amino acids pathway. While deletions of most the genes that encode for proteins involved in the de novo biosynthesis of sulfurcontaining amino acids, result in auxotrophy for methionine or cysteine, ∆∆met32 strain does not exhibit this phenotype (Blaiseau et al., 1997). This key distinction is attributed to the functional redundancy provided by the paralogous gene, met31 (Blaiseau et al., 1997). Crucially, given that the deletion of the central transcriptional regulator, met4, results in cysteine/methionine auxotrophy, the use of the ∆∆met32 strain provides an essential, viable experimental model for investigating the role of sulfur metabolism during pseudohyphal differentiation in *S. cerevisiae*.

(2) Comparison of consistent gene and protein expression data (Met30, Met4, Met32) across all relevant figures and analyses would strengthen the mechanistic connection in a better way. Some data that might help connect the sections is not included; please see the public review for more details.

We thank the reviewer for this valuable input, which helps us to clarify the scope of our transcriptomic analysis. Our decision to focus our RT-qPCR experiments on downstream targets, while excluding Met4 and Met30 from the RT-qPCR analysis, is based on their known regulatory mechanisms. Met4 activity is predominantly regulated by post-translational ubiquitination by the SCFMet30 complex followed by its degradation (Rouillon et al., 2000; Shrivastava et al., 2021; Smothers et al., 2000) while Met30 activity is primarily regulated by its auto-degradation or its dissociation from the SCFMet30 complex (Lauinger et al., 2024; Smothers et al., 2000; Yen et al., 2005). Consistent with this, our RNA-Seq results indicate that neither met4 nor met30 transcripts are differentially expressed, in response to 2DG addition. For all our RT-qPCR analysis in *S. cerevisiae* and *C. albicans*, we have consistently used the same set of sulfur metabolism genes and these include met32, met3, met5, met10 and met17. Our data on protein expression analysis of Met30 in *S. cerevisiae* (Fig. 3J) confirms that Met30 expression is not differentially regulated in the presence of 2DG, effectively eliminating changes in synthesis or SCFMet30 proteasomal degradation as the dominant regulatory mechanism.

(3) Suggested to include metabolomic profiling (cysteine, methionine, and intermediate metabolites) to substantiate the proposed metabolic flux between glycolysis and sulfur metabolism.

We thank the reviewer for this valuable input. Our pseudohyphal/hyphal differentiation assays show that the defects induced by glycolytic perturbation is fully rescued by the exogenous supplementation of sulfur-containing amino acids, cysteine or methionine. Since these data conclusively demonstrate that the primary metabolic limitation caused by the perturbation of glycolysis, which leads to filamentation defects, is sulfur metabolism, we posit that performing comprehensive metabolic profiling would primarily reconfirm the aforesaid results. We believe that our in vitro and in vivo sulfur add-back experiments sufficiently substantiate the novel regulatory metabolic link between glycolysis and sulfur-metabolism.

(4) Data on the effects of Met30 deletion on cell growth are currently not included, and relevant controls should be included to ensure observed phenotypes are not due to general growth defects.

We are grateful to the reviewer for this constructive feedback. To address the potential impact of met30 deletion on cell growth, we have included new data (Suppl. Fig. 4A) demonstrating that the deletion of a single copy of met30 in diploid *S. cerevisiae* does not compromise overall growth under nitrogen-limiting conditions as the ∆met30 strain grows similar to the wild-type strain.

(5) Expanding RT-qPCR and data from transcriptomic analyses to include sulfur metabolism genes and key cAMP pathway genes to confirm the proposed cAMP-independent mechanism during virulence characterization is necessary.

We thank the reviewer for this valuable feedback. The transcriptional analysis of the sulfur metabolism genes in the presence of 2DG and the ∆∆pfk1 strain is shown in Figures 4D and 4I.

In order to confirm that glycolysis is critical for fungal morphogenesis in a cAMP-PKA-independent manner under nitrogen-limiting conditions in *C. albicans*, we performed cAMP add-back assays. Interestingly, corroborating our *S. cerevisiae* data, the exogenous addition of cAMP failed to rescue hyphal differentiation defect caused by the perturbation of glycolysis through 2DG addition or by the deletion of the pfk1 gene, under nitrogen-limiting condition in C. albicans. This data is now included in Suppl. Fig. 5B.

(6) Enhancing the introduction and discussion by providing a clearer rationale for gene selection and more detailed references to established pathways (cAMP-PKA, MAPK, Snf1/HXT regulation, gpa2 involvement) is needed to reinstate the hypothesis.

We thank the reviewer for this valuable feedback. We have incorporated these changes in our revised manuscript.

(7) Reducing redundancy in the text and improving figure consistency, particularly by ensuring that the gene sets depicted in Figure 6 are represented across all datasets, would strengthen the interconnections among sections.

We thank the reviewer for this valuable feedback. We have incorporated these changes in our revised manuscript.

References

Barford JP, Hall RJ. 1979. An examination of the crabtree effect in *Saccharomyces cerevisiae*: The role of respiratory adaptation. J Gen Microbiol. https://doi.org/10.1099/00221287-114-2-267

Blaiseau, P. L., & Thomas, D. (1998). Multiple transcriptional activation complexes tether the yeast activator Met4 to DNA. The EMBO journal, 17(21), 6327–6336. https://doi.org/10.1093/emboj/17.21.6327

Chebaro, Y., Lorenz, M., Fa, A., Zheng, R., & Gustin, M. (2017). Adaptation of Candida albicans to Reactive Sulfur Species. Genetics, 206(1), 151–162. https://doi.org/10.1534/genetics.116.199679

De Deken R. H. (1966). The Crabtree effect: a regulatory system in yeast. Journal of general microbiology, 44(2), 149–156. https://doi.org/10.1099/00221287-44-2-149

Düring-Olsen, L., Regenberg, B., Gjermansen, C., Kielland-Brandt, M. C., & Hansen, J. (1999). Cysteine uptake by *Saccharomyces cerevisiae* is accomplished by multiple permeases. Current genetics, 35(6), 609–617. https://doi.org/10.1007/s002940050459

Gancedo J. M. (2001). Control of pseudohyphae formation in *Saccharomyces cerevisiae*. FEMS microbiology reviews, 25(1), 107–123. https://doi.org/10.1111/j.1574-6976.2001.tb00573.x

Gimeno, C. J., Ljungdahl, P. O., Styles, C. A., & Fink, G. R. (1992). Unipolar cell divisions in the yeast *S. cerevisiae* lead to filamentous growth: regulation by starvation and RAS. Cell, 68(6), 1077–1090. https://doi.org/10.1016/0092-8674(92)90079-r

Huang, C. W., Walker, M. E., Fedrizzi, B., Gardner, R. C., & Jiranek, V. (2017). Yeast genes involved in regulating cysteine uptake affect production of hydrogen sulfide from cysteine during fermentation. FEMS yeast research, 17(5), 10.1093/femsyr/fox046. https://doi.org/10.1093/femsyr/fox046

Kosugi, A., Koizumi, Y., Yanagida, F., & Udaka, S. (2001). MUP1, high affinity methionine permease, is involved in cysteine uptake by *Saccharomyces cerevisiae*. Bioscience, biotechnology, and biochemistry, 65(3), 728–731. https://doi.org/10.1271/bbb.65.728

Kraidlova, L., Schrevens, S., Tournu, H., Van Zeebroeck, G., Sychrova, H., & Van Dijck, P. (2016). Characterization of the Candida albicans Amino Acid Permease Family: Gap2 Is the Only General Amino Acid Permease and Gap4 Is an S-Adenosylmethionine (SAM) Transporter Required for SAM-Induced Morphogenesis. mSphere, 1(6), e00284-16. https://doi.org/10.1128/mSphere.00284-16

Lauinger, L., Andronicos, A., Flick, K., Yu, C., Durairaj, G., Huang, L., & Kaiser, P. (2024). Cadmium binding by the F-box domain induces p97-mediated SCF complex disassembly to activate stress response programs. Nature communications, 15(1), 3894. https://doi.org/10.1038/s41467-024-48184-6

Lombardi, L., Salzberg, L. I., Cinnéide, E. Ó., O'Brien, C., Morio, F., Turner, S. A., Byrne, K. P., & Butler, G. (2024). Alternative sulphur metabolism in the fungal pathogen Candida parapsilosis. Nature communications, 15(1), 9190. https://doi.org/10.1038/s41467-024-53442-8

Menant, A., Barbey, R., & Thomas, D. (2006). Substrate-mediated remodeling of methionine transport by multiple ubiquitin-dependent mechanisms in yeast cells. The EMBO journal, 25(19), 4436–4447. https://doi.org/10.1038/sj.emboj.7601330

Ralser, M., Wamelink, M. M., Kowald, A., Gerisch, B., Heeren, G., Struys, E. A., Klipp, E., Jakobs, C., Breitenbach, M., Lehrach, H., & Krobitsch, S. (2007). Dynamic rerouting of the carbohydrate flux is key to counteracting oxidative stress. Journal of biology, 6(4), 10. https://doi.org/10.1186/jbiol61

Rouillon, A., Barbey, R., Patton, E. E., Tyers, M., & Thomas, D. (2000). Feedback-regulated degradation of the transcriptional activator Met4 is triggered by the SCF(Met30)complex. The EMBO journal, 19(2), 282–294. https://doi.org/10.1093/emboj/19.2.282

Schrevens, S., Van Zeebroeck, G., Riedelberger, M., Tournu, H., Kuchler, K., & Van Dijck, P. (2018). Methionine is required for cAMP-PKA-mediated morphogenesis and virulence of Candida albicans. Molecular microbiology, 108(3), 258–275. https://doi.org/10.1111/mmi.13933

Shee, S., Singh, S., Tripathi, A., Thakur, C., Kumar T, A., Das, M., Yadav, V., Kohli, S., Rajmani, R. S., Chandra, N., Chakrapani, H., Drlica, K., & Singh, A. (2022). Moxifloxacin-Mediated Killing of *Mycobacterium tuberculosis* Involves Respiratory Downshift, Reductive Stress, and Accumulation of Reactive Oxygen Species. Antimicrobial agents and chemotherapy, 66(9), e0059222. https://doi.org/10.1128/aac.00592-22

Shrivastava, M., Feng, J., Coles, M., Clark, B., Islam, A., Dumeaux, V., & Whiteway, M. (2021). Modulation of the complex regulatory network for methionine biosynthesis in fungi. Genetics, 217(2), iyaa049. https://doi.org/10.1093/genetics/iyaa049

Smothers, D. B., Kozubowski, L., Dixon, C., Goebl, M. G., & Mathias, N. (2000). The abundance of Met30p limits SCF(Met30p) complex activity and is regulated by methionine availability. Molecular and cellular biology, 20(21), 7845–7852. https://doi.org/10.1128/MCB.20.21.7845-7852.2000

Thomas, D., & Surdin-Kerjan, Y. (1997). Metabolism of sulfur amino acids in *Saccharomyces cerevisiae*. Microbiology and molecular biology reviews : MMBR, 61(4), 503–532. https://doi.org/10.1128/mmbr.61.4.503532.1997

Yadav, A. K., & Bachhawat, A. K. (2011). CgCYN1, a plasma membrane cystine-specific transporter of Candida glabrata with orthologues prevalent among pathogenic yeast and fungi. The Journal of biological chemistry, 286(22), 19714–19723. https://doi.org/10.1074/jbc.M111.240648

Yen, J. L., Su, N. Y., & Kaiser, P. (2005). The yeast ubiquitin ligase SCFMet30 regulates heavy metal response. Molecular biology of the cell, 16(4), 1872–1882. https://doi.org/10.1091/mbc.e04-12-1130